# The larva and adult of *Helicoverpa armigera* use differential gustatory receptors to sense sucrose

**Shuai-Shuai Zhang**[1,2†], **Pei-Chao Wang**[1,2†], **Chao Ning**[1,2], **Ke Yang**[1,2], **Guo-Cheng Li**[1,2], **Lin-Lin Cao**[1,2], **Ling-Qiao Huang**[1], **Chen-Zhu Wang**[1,2*]

[1]State Key Laboratory of Integrated Management of Pest Insects and Rodents, Institute of Zoology, Chinese Academy of Sciences, Beijing, China; [2]Chinese Academy of Sciences Center for Excellence in Biotic Interactions, University of Chinese Academy of Sciences, Beijing, China

**\*For correspondence:**
czwang@ioz.ac.cn

[†]These authors contributed equally to this work

**Competing interest:** The authors declare that no competing interests exist.

**Abstract** Almost all herbivorous insects feed on plants and use sucrose as a feeding stimulant, but the molecular basis of their sucrose reception remains unclear. *Helicoverpa armigera* as a notorious crop pest worldwide mainly feeds on reproductive organs of many plant species in the larval stage, and its adult draws nectar. In this study, we determined that the sucrose sensory neurons located in the contact chemosensilla on larval maxillary galea were 100–1000 times more sensitive to sucrose than those on adult antennae, tarsi, and proboscis. Using the *Xenopus* expression system, we discovered that Gr10 highly expressed in the larval sensilla was specifically tuned to sucrose, while Gr6 highly expressed in the adult sensilla responded to fucose, sucrose and fructose. Moreover, using CRISPR/Cas9, we revealed that Gr10 was mainly used by larvae to detect lower sucrose, while Gr6 was primarily used by adults to detect higher sucrose and other saccharides, which results in differences in selectivity and sensitivity between larval and adult sugar sensory neurons. Our results demonstrate the sugar receptors in this moth are evolved to adapt toward the larval and adult foods with different types and amounts of sugar, and fill in a gap in sweet taste of animals.

## eLife assessment

This **important** study identifies the gustatory receptors for sugar sensing in the larval and adult forms of the cotton bollworm, which is responsible for the destruction of many food crops worldwide. The authors find that the larval and adult forms utilise different receptors to sense sugars. The data are **convincing** and will be of interest neuroscientists working in sensory coding of sugars and to the pest management field.

## Introduction

Over 400,000 species of phytophagous insects live on approximately 300,000 species of vascular plants worldwide (*Schoonhoven et al., 2005*). Carbohydrates produced by plant photosynthesis are important nutrients for phytophagous insects. The disaccharide sucrose, the major sugar in plants, is the principal transport form of photoassimilates in higher plants. Sucrose varies considerably in concentration in green leaves, fruits, and flower nectar where it occurs together with its constituent monosaccharides fructose and glucose (*Bernays and Chapman, 1994*). Sucrose, fructose, and glucose are the powerful phagostimulants for most phytophagous insects studied (*Bernays and Chapman, 1994*; *Blaney and Simmonds, 1988*; *Chapman, 2003*; *Schoonhoven et al., 2005*).

The gustatory sensory neurons (GSNs) in one or more mouthpart sensilla of phytophagous insects respond to sugars, usually including sucrose (*Chapman, 2003*; *Dethier, 1973*). In Lepidoptera, caterpillars often have sugar-sensitive GSNs in lateral or medial sensilla styloconica on the maxillary galea (*Glendinning et al., 2009*; *Martin and Shields, 2012*; *Schoonhoven and Loon, 2002*), while adults also have sugar-sensitive GSNs in contact chemosensilla of the antennae (*Jørgensen et al., 2007*; *Popescu et al., 2013*), tarsi (*Schnuch and Seebauer, 1998*; *Zhang et al., 2010*), and proboscis (*Blaney and Simmonds, 1988*; *Blaney and Simmonds, 1990*; *Faucheux, 2013*). The molecular basis of sugar reception in Lepidopteran insects remains unclear, although the sweet taste receptors in mammals and *Drosophila melanogaster* have been extensively studied (*Liman et al., 2014*; *Yarmolinsky et al., 2009*). In mammals, the combination of two taste receptors T1R2 and T1R3 recognizes sugars including sucrose (*Damak et al., 2003*; *Li et al., 2002*; *Nelson et al., 2001*; *Zhao et al., 2003*). In *Drosophila*, multiple gustatory receptors (GRs), including DmGr5a, DmGr61a, and DmGr64a–f clusters, are required for the response of adults to multiple sugars (*Dahanukar et al., 2001*; *Dahanukar et al., 2007*; *Jiao et al., 2007*; *Jiao et al., 2008*; *Slone et al., 2007*); DmGr43a specifically responding to fructose is the major larval sugar receptor, but it is only expressed in internal organs and some parts of the brain (*Mishra et al., 2013*; *Miyamoto et al., 2012*). No external sugar-sensing mechanism in *Drosophila* larvae has yet been characterized.

The cotton bollworm *Helicoverpa armigera* (Noctuidae, Lepidoptera) is a worldwide agricultural pest, it causes over \$3 billion in annual global economic loss (*Pearce et al., 2017*). The larvae of *H. armigera* mainly feeds on young leaves, flower buds, and fruits of host plants; the adults mainly drink nectar of plants. Larval and adult diets vary dramatically in the variety and concentration of sugars. Sucrose is the major sugar in plant leaves and reproductive organs, whereas flower nectar mainly contains sucrose, fructose, and glucose (*Nicolson, 1998*). Previous studies have shown that one sugar-sensitive GSN in *H. armigera* is located in each of two lateral sensilla styloconica in the larva (*Tang et al., 2000*; *Zhang et al., 2013*), and many sugar-sensitive GSNs are distributed in the numerous contact chemosensilla of antennae (*Jiang et al., 2015*), tarsi (*Blaney and Simmonds, 1990*; *Zhang et al., 2010*), and proboscis (*Blaney and Simmonds, 1988*) in the adult.

To understand the molecular basis of sugar reception in these neurons, we first compared electrophysiological and behavioral responses of *H. armigera* larvae and adults. Second, we analyzed the expression patterns of nine candidate sugar GR genes in larval and adult taste organs. Third, we carried out functional analysis of these GR genes using the ectopic expression system, and revealed the response profiles of *Xenopus oocytes* expressing Gr10, Gr6, and both of them. Finally, we individually knocked out the two GR genes and obtained two homozygous mutants of *H. armigera* by CRISPR/Cas9, and then detected changes in the electrophysiological and behavioral responses of the mutant larvae and adults to sugars. Taking all the results together, we demonstrate that larval and adult *H. armigera* used different GRs to detect sucrose in food.

## Results

### Electrophysiological responses of contact chemosensilla to sugars

We first investigated the electrophysiological responses of the lateral and medial sensilla styloconica on the larval maxillary galea to eight sugars. These sugars were chosen because they are mostly found in host-plants of *H. armigera* or are representative in the structure and source of sugars. The lateral sensilla styloconica showed significant unicellular responses to 10 mM sucrose and fucose, with the firing rates of 169±8.57 and 43±9.78 spikes/s, respectively (*Figure 1A*, *Figure 1—figure supplement 1A*). The threshold concentrations for their responses to sucrose and fucose were 0.1 mM and 10 mM, respectively (*Figure 1B*, *Figure 1—figure supplement 1B*). The medial sensilla styloconica had a weak response to xylose (*Figure 1—figure supplement 1C*). We also compared the responses of the lateral sensilla styloconica to 0.05 mM sucrose, 5 mM fucose, and their mixture with the same concentrations, and found that the mixture induced higher responses with only one amplitude spikes and a firing rate of 71±12.36 spikes/s, which was higher than the firing rates induced by 0.05 mM sucrose (39±6.37 spikes/s) and 5 mM fucose (36±8.87 spikes/s; *Figure 1—figure supplement 1D*), confirming that the same cell responded to sucrose and fucose. We conclude that there is only one GSN sensitive to sucrose and fucose in each maxillary galea of larvae.

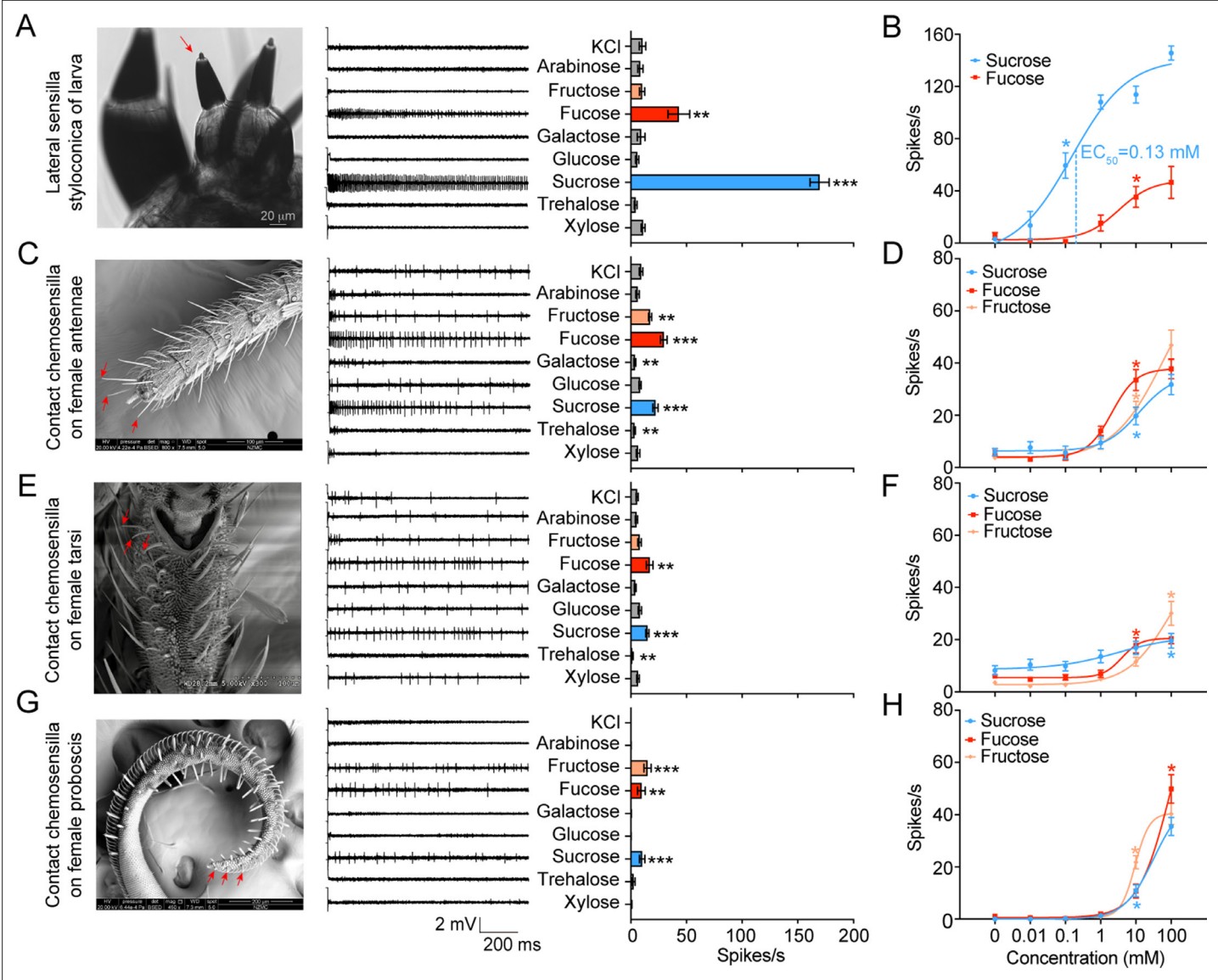

**Figure 1.** Electrophysiological responses of larval and adult contact chemosensilla in *Helicoverpa armigera* to sugars. (**A**) The maxilla of the larva (left), the representative spike traces of the responses of lateral sensilla styloconica on larval maxillary galea to eight sugars at 10 mM (middle), and quantifications of the firing rates (right) (n=13). The red arrow marks the recorded lateral sensillum styloconicum. Scale bar represents 20 µm. (**B**) Dose–responses of the lateral sensilla styloconica on larval maxillary galea to sucrose and fucose (sucrose: n=8; fucose: n=6). (**C**) The antenna terminal of the female (left), the representative spike traces of the responses of the contact chemosensilla on female antennae to eight sugars at 10 mM (middle), and quantifications of the firing rates (right) (n=21). The red arrows mark the recorded contact chemosensilla. Scale bar represents 100 µm. (**D**) Dose–responses of the contact chemosensilla on female antennae to sucrose, fucose, and fructose (n=21). (**E**) The fifth tarsomere of the female (left) (***Zhang et al., 2010***), the representative spike traces of the responses of contact chemosensilla on female tarsi to eight sugars at 10 mM (middle), and quantifications of the firing rates (right) (n=15). The red arrows mark the recorded contact chemosensilla. Scale bar represents 100 µm. (**F**) Dose–responses of the contact chemosensilla on female tarsi to sucrose, fucose, and fructose (n=21). (**G**) The proboscis terminal of the female (left), the representative spike traces of the responses of contact chemosensilla on female proboscis to eight sugars at 10 mM (middle), and quantifications of the firing rates (right) (n=21). The red arrows mark the recorded contact chemosensilla. Scale bar represents 200 µm. (**H**) Dose–responses to sucrose (n=21), fucose (n=21), and fructose (fructose 0 mM, n=18; fructose 0.01 mM-100 mM, n=21) of the contact chemosensilla on the female proboscis. (A to H) Data are mean ± SEM; * $p<0.05$; ** $p<0.01$; *** $p<0.001$. (**A, C, E, G**) Data were analyzed by independent-samples *t* test (compared with control). (**B, D, F, H**) Data were analyzed by one-way ANOVA with Tukey's HSD test.

© 2010, Journal of Experimental Biology. Figure 1E is taken from Figure 1A in ***Zhang et al., 2010***, Journal of Experimental Biology. It is not covered by the CC-BY 4.0 license and further reproduction of this panel would need permission from the copyright holder.

The online version of this article includes the following source data and figure supplement(s) for figure 1:

*Figure 1 continued on next page*

*Figure 1 continued*

**Source data 1.** Electrophysiological responses of larval and adult contact chemosensilla in *Helicoverpa armigera* to sugars.

**Figure supplement 1.** Electrophysiological responses of contact chemosensilla in *Helicoverpa armigera* to sugars.

**Figure supplement 1—source data 1.** Electrophysiological responses of contact chemosensilla in *Helicoverpa armigera* to sugars.

We also studied the responses of the contact chemosensilla on the top areas of the antennae, foreleg tarsi, and proboscis of female adults to eight sugars at 10 mM and the related dose responses (*Figure 1C–H*). The antennal sensilla responded to sucrose, fucose, and fructose positively (*Figure 1C*), the tarsal sensilla only responded to sucrose and fucose (*Figure 1E*), and the proboscis sensilla responded to sucrose, fucose, and fructose (*Figure 1G*), but their firing rates were between 9 and 22 spikes/s. All the evoked spike traces had one amplitude, indicating that they were single cell responses (*Figure 1—figure supplement 1A*). The dose–response curves showed that the threshold concentrations of sucrose, fucose and fructose were all 10 mM for the antennal sensilla (*Figure 1D*, *Figure 1—figure supplement 1B*), 100 mM, 10 mM and 100 mM for the tarsal sensilla (*Figure 1F*, *Figure 1—figure supplement 1B*), and 10 mM, 100 mM and 10 mM for the proboscis sensilla, respectively (*Figure 1H*, *Figure 1—figure supplement 1B*).

In short, both larvae and adults of *H. armigera* had sugar GSNs, but their response profiles, intensity and sensitivity differed distinctively. The sugar GSNs in lateral sensilla styloconica of larvae responded strongly to sucrose with high sensitivity, and weakly to fucose; while the sugar GSNs of adults responded to sucrose, fucose, and fructose (except for those on tarsi) with lower intensity and sensitivity. The GSNs of larvae were 100–1,000 times more sensitive to sucrose than those of adults.

## Behavioral responses of larvae and adults to sugars

To determine how the taste inputs of various sugars cause behavioral responses in insects, the two-choice test was used to examine the feeding behavior of the 5th instar larvae of *H. armigera* to eight sugars at 10 mM. Sucrose and fructose stimulated larval feeding, and their threshold concentrations were 10 mM (*Figure 2A–B*, *Figure 2—figure supplement 1*); other tested sugars had no effect on larval feeding (*Figure 2A*).

We also studied the proboscis extension reflex (PER) induced by sugar stimulation in the top areas of antennae and fore leg tarsi of virgin female adults. In consideration of the sugar content in nectar and the response intensity of the contact chemosensilla in adults, the test concentration of sugars was set at 100 mM. The results showed that the PER percentage reached 64 ± 4.35%, 44 ± 5.96% and 20 ± 1.92% when the antennal sensilla were stimulated with sucrose, fucose and fructose (*Figure 2C*), while the PER percentage reached 53 ± 14.08%, 45 ± 4.44% and 36 ± 10.19% when the tarsal sensilla were stimulated, respectively (*Figure 2D*). Stimulation of the antennal or tarsal sensilla by other sugars did not induce changes in the PER percentage compared with stimulation of water (*Figure 2C–D*). The dose–response curves showed that the threshold concentrations of sucrose, fucose, and fructose for the antennal sensilla were 100 mM, 100 mM, and 1000 mM (*Figure 2E*), those for the tarsal sensilla were 10 mM, 10 mM, and 100 mM, respectively (*Figure 2F*).

## Phylogenetic tree and expression patterns of putative sugar gustatory receptors

On the basis of understanding the electrophysiological and behavioral responses of larvae and adults of *H. armigera* to various sugars, we aimed to reveal the molecular mechanisms of sugar reception in the contact chemosensilla of *H. armigera*. We first analyzed the putative sugar gustatory receptor genes based on the genome data of *H. armigera* (*Pearce et al., 2017*), the reported gene sequences of sugar GRs in *H. armigera* and their phylogenetic relationship of *D. melanogaster* sugar gustatory receptors (*Jiang et al., 2015*; *Xu et al., 2012*). All nine putative sugar GR genes in *H. armigera*, *Gr4–12* were validated, and their full-length cDNA sequences were cloned (The GenBank accession number is provided in *Supplementary file 1*). The lengths of the amino acid sequences of the sugar GRs were consistent with the lengths of the corresponding reported sequences (the similarity was 97–99%), except for Gr8, Gr9, and Gr10. Gr8 had two more amino acids at the C-terminal than the reported by *Xu et al., 2017* with 99% sequence similarity (*Xu et al., 2017*). Gr9 had 9 more amino acids at the N-terminal than the reported in the *H. armigera* genome study with 99% sequence

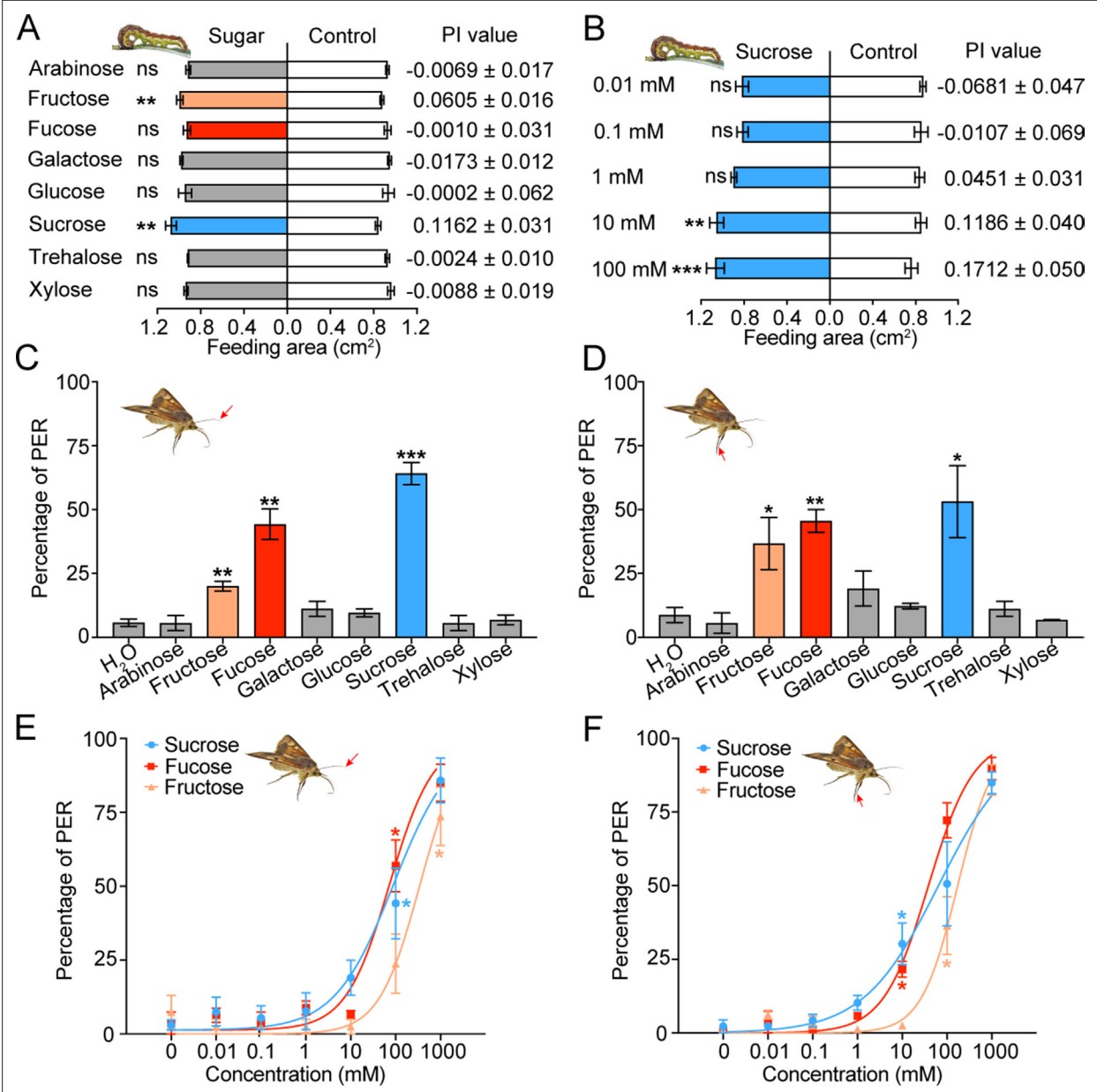

**Figure 2.** Behavioral responses of *Helicoverpa armigera* larvae and adults to sugars. (**A**) Feeding responses and the preference index (PI) value of 5th instar larvae to eight sugars painted on the cabbage leaf discs at 10 mM in two-choice tests. ** $p<0.01$; ns indicates no significance, $p \geq 0.05$ (paired *t* test, n=20). (**B**) Feeding responses and the PI value of 5th instar larvae to different concentrations of sucrose painted on the cabbage leaf discs in two-choice tests. ** $p<0.01$, *** $p<0.001$; ns indicates no significance, $p \geq 0.05$ (paired *t* test, n=20). (**C**) Proboscis extension reflex (PER) in adult females upon antennal stimulation by eight sugars at 100 mM. ** $p<0.01$; *** $p<0.001$ (independent-samples *t* test compared with control, n=3). (**D**) PER in adult females upon tarsal stimulation by eight sugars at 100 mM. * $p<0.05$; ** $p<0.01$ (independent-samples *t* test, n=3). (**E**) PER in adult females upon antennal stimulation by different concentrations of sucrose, fucose, and fructose. * $p<0.05$ (one-way ANOVA with Tukey's HSD test, n=3). (**F**) PER in adult females upon tarsal stimulation by different concentrations of sucrose, fucose, and fructose (n=3). * $p<0.05$ (one-way ANOVA with Tukey's HSD test, n=3). (A to F) Data are mean ± SEM. The red arrow indicates the stimulating site.

The online version of this article includes the following source data and figure supplement(s) for figure 2:

**Source data 1.** Behavioral responses of *Helicoverpa armigera* larvae and adults to sugars.

**Figure supplement 1.** Feeding responses and the PI value of 5th instar larvae of *Helicoverpa armigera* to different concentrations of fructose painted on the cabbage leaf discs in two-choice tests.

**Figure supplement 1—source data 1.** Feeding area of *H. armigera* 5th-instar larvae in two-choice tests to fructose at different concentrations.

similarity (*Pearce et al., 2017*), while the sequence length of Gr9 was exactly the same as Gr4 in *Jiang et al., 2015* with 99% sequence similarity (*Jiang et al., 2015*). Gr10 had 20 more amino acids than the sequence reported in the *H. armigera* genome study (*Pearce et al., 2017*) and 186 more amino acids than the sequence reported by *Xu et al., 2017* (*Xu et al., 2017*), with 97% and 99% sequence similarities, respectively.

We established a phylogenetic tree using sugar GR genes of nine insect species, including *D. melanogaster*, *Aedes aegypti*, and *Anopheles gambiae* in Diptera; *Apis mellifera* in Hymenoptera; and *Danaus plexippus*, *Heliconius melpomene*, *Bombyx mori*, *Plutella xylostella*, and *H. armigera* in Lepidoptera (*Figure 3A*). The phylogenetic analysis showed that candidate sugar GRs of these nine insect species grouped within two clades, the sugar GR clade and the fructose GR clade. All the putative sugar GRs of *H. armigera* were in the sugar GR clade except for HarmGr9, which was in the fructose GR clade (*Figure 3A*).

To determine the expression patterns of nine putative sugar GR genes in larval and adult taste organs, we measured their expression levels in the larval maxillary galea, and the antennae, foreleg tarsi, and proboscis in virgin female and male adults by real-time quantitative PCR (qRT-PCR) (*Figure 3B–E*, *Figure 3—figure supplement 1*). The results showed that the expression patterns of sugar GRs in the same organ of male and female adults were nearly the same (*Figure 3C–E*, *Figure 3—figure supplement 1*). The average expression level of all sugar GRs in the larval maxillary galea was lower than that in the adult organs. In each taste organ, the expression levels of 1–2 sugar GRs were much higher than those of other sugar GRs, and these highly expressed GRs were significantly different between larvae and adults (*Figure 3B–E*, *Figure 3—figure supplement 1*). Among all sugar GRs, *Gr10* was highly expressed in the larval maxillary galea, whereas *Gr6* was highly expressed in the antennae, tarsi, and proboscis of both male and female adults. *Gr5* expression levels greatly varied; they were highest in the proboscis followed by the tarsi, but very low in the antennae and the maxillary galea of the larvae (*Figure 3B–E*, *Figure 3—figure supplement 1*).

## Functional analysis of putative sugar gustatory receptors by ectopic expression

Of all the nine putative sugar GRs in *H. armigera* mentioned above, only Gr9, originally named as Gr4, has been previous identified as a fructose receptor (*Jiang et al., 2015*). To explore the functions of the other eight GRs, we first investigated the inward current changes in *Xenopus* oocytes expressing sugar GRs individually in response to the stimulation of 11 sugars at 100 mM using two-electrode voltage-clamp. The results showed that the oocytes expressing Gr10 only responded to sucrose with a current value of 4.10±0.53 µA (*Figure 4A–B*); the oocytes expressing Gr6 showed responses to fucose and sucrose with a current value of 1.59±0.08 µA and 0.24±0.03 µA, respectively (*Figure 4A–B*). The oocytes expressing remaining six sugar GR genes (Gr4, Gr5, Gr7, Gr8, Gr11, and Gr12) singly had no response to the tested sugars (*Figure 4—figure supplement 1A–B*). The dose–response curves showed that the threshold concentration of sucrose for the oocytes expressing Gr10 was 100 mM (*Figure 4C–D*), whereas the threshold concentrations of sucrose and fucose for the oocytes expressing Gr6 were 250 mM and 100 mM, respectively, and 500 mM fructose could also induce an inward current (*Figure 4C–D*).

Given that there was only one sugar GSN responding to sucrose and fucose in each larval maxillary galea, we speculated that Gr10 should be expressed together with Gr6 in this cell, although their expressing levels were different. Therefore, we examined the response profile of the oocytes expressing Gr10 together with Gr6. The results showed that the oocytes co-expressing Gr10 and Gr6 significantly responded to sucrose with a current value of 1.89±0.58 µA at 100 mM, a threshold concentration, and had a weak response to fucose with a threshold concentration of 250 mM (*Figure 4*).

Since Gr5 and Gr6 were highly expressed in the proboscis and tarsi (*Figure 3D–E*, *Figure 3—figure supplement 1*), we suspected that Gr5 and Gr6 might be expressed in the same cells, and then tested the response profile of their co-expression in oocytes. Similar to the oocytes expressing Gr6, the oocytes expressing Gr5 and Gr6 had responses to 100 mM fucose with a current value of 0.11±0.015 µA and to 100 mM sucrose with a current value of 0.03±0.002 µA (*Figure 4—figure supplement 1B–C*).

In summary, we find that oocytes expressing single Gr10 or Gr6 confer sensitivity to one or more sugars, indicating their direct roles in ligand recognition. Gr10 is tuned to sucrose specifically, Gr6 is

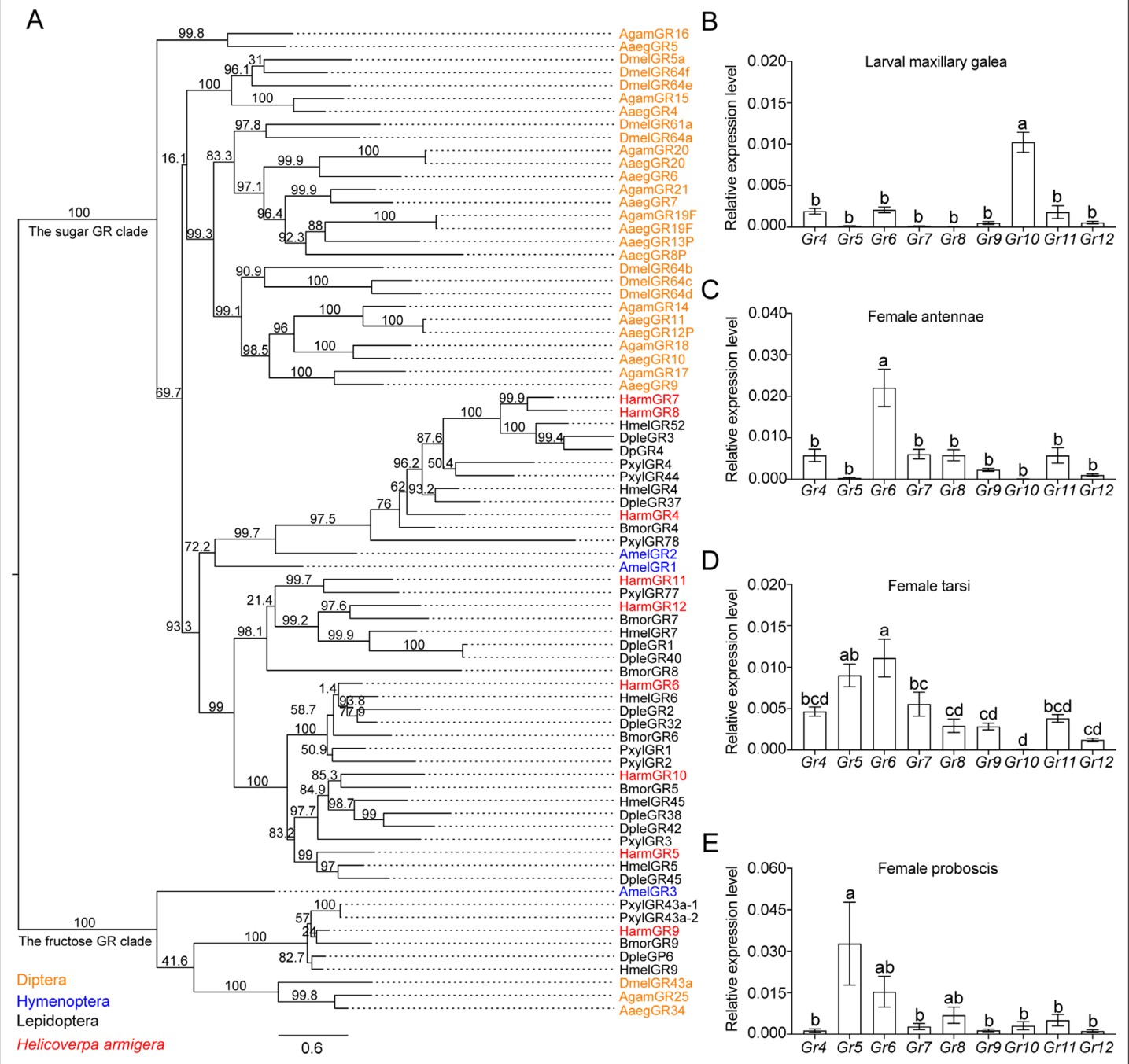

**Figure 3.** The phylogenetic relationship and the expression level of sugar GRs in larval maxilla and female adult antennae, tarsi and proboscis of *Helicoverpa armigera*. (**A**) The phylogenetic tree of insect sugar GRs. Diptera (orange): Aaeg, *Aedes aegypti*; Agam, A*nopheles gambiae*; Dmel, *Drosophila melanogaster*. Hymenoptera (blue): Am, *Apis mellifera*. Lepidoptera (black): Bmor, *Bombyx mori*; Dple, *Danaus plexippus*; Hmel, *Heliconius melpomene*; Pxyl, *Plutella xylostella*. Harm (red), *Helicoverpa armigera*. Numbers above branches indicate ultrafast bootstrap approximation (UFBoot). (**B**) Relative expression levels of sugar GRs in the maxillary galea of 5th instar larvae of *H. armigera* determined by qRT-PCR. (**C**) Relative expression levels of sugar GRs in the female adult antennae. (**D**) Relative expression levels of sugar GRs in female adult tarsi. (**E**) Relative expression levels of sugar GRs in female adult proboscis. (B to E) Data are mean ± SEM. One-way ANOVA was used, and different letters labeled indicate significant difference (Tukey's HSD test, p<0.05, n=3).

The online version of this article includes the following source data and figure supplement(s) for figure 3:

**Source data 1.** The expression levels of sugar GRs in larval maxilla and female adult antennae, tarsi and proboscis of *Helicoverpa armigera*.

**Figure supplement 1.** Expression patterns of sugar GRs in taste organs of *Helicoverpa armigera* male adults.

**Figure supplement 1—source data 1.** Expression patterns of sugar GRs in taste organs of *Helicoverpa armigera* male adults.

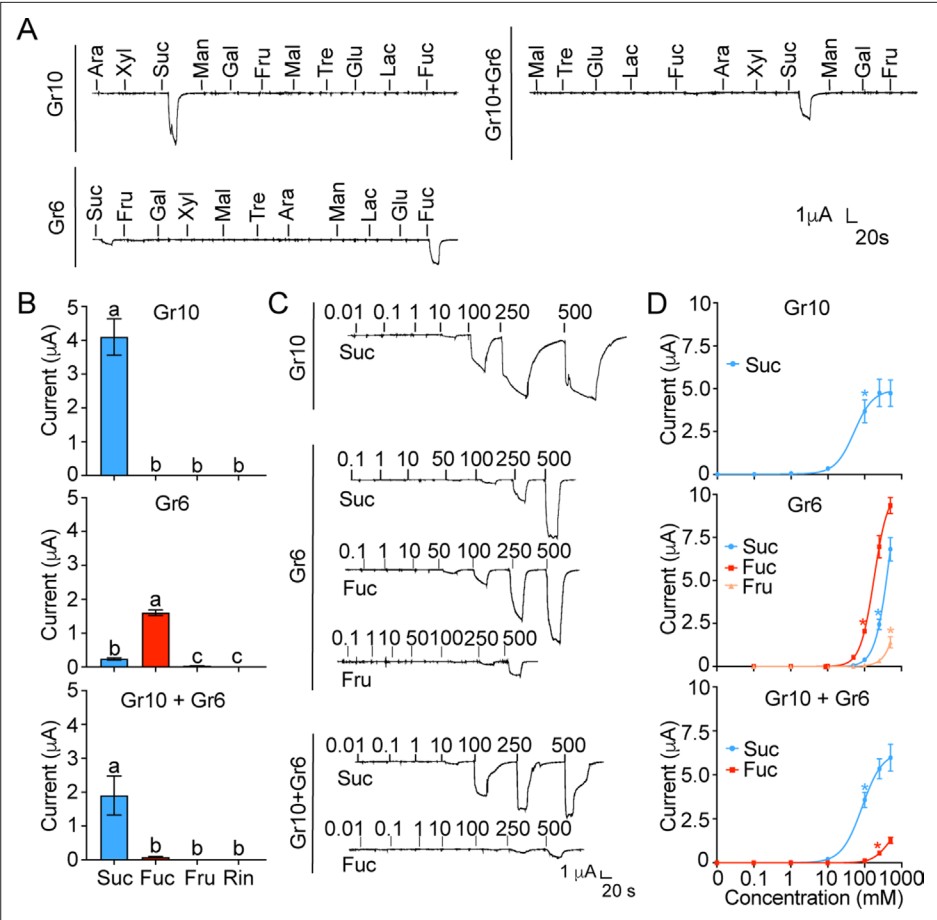

**Figure 4.** The inward current responses of the *Xenopus* oocytes expressing sugar GRs of *Helicoverpa armigera* to sugars. (**A**) The representative traces of the oocytes expressing Gr10, Gr6, and Gr10 +Gr6 to 11 sugars at 100 mM. (**B**) The responses of the oocytes expressing Gr10, Gr6, and Gr10 +Gr6 to sugars at 100 mM (Gr10, n=14. Gr6, n=20. Gr10 +Gr6, n=7). (**C**) The representative trace of the oocytes expressing Gr10 to sucrose (the upper), the representative trace of the oocytes expressing Gr6 to sucrose, fucose, and fructose (the middle), the representative trace of the oocytes expressing Gr10 +Gr6 to sucrose and fucose (the lower). (**D**) The dose-responses of the oocytes expressing Gr10 to sucrose (the upper, n=13); the dose-responses of the oocytes expressing Gr6 to sucrose, fucose, and fructose (the middle, sucrose: n=6; fucose: n=6; fructose: n=3); the dose-responses of the oocytes expressing Gr10 +Gr6 to sucrose and fucose (the lower, sucrose: n=4; fucose: n=3). (**B and D**) Data are mean ± SEM, and were analyzed by one-way ANOVA with Tukey's HSD test (p<0.05). Different letters labeled indicate significant differences, * p<0.05. (A to D) Ara, arabinose; Fru, fructose; Fuc, fucose; Gal, galactose; Glu, glucose; Lac, lactose; Mal, maltose; Man, mannose; Suc, sucrose; Tre, trehalose; Xyl, xylose; Rin, Ringer solution.

The online version of this article includes the following source data and figure supplement(s) for figure 4:

**Source data 1.** The inward current responses of the *Xenopus* oocytes expressing sugar GRs of *Helicoverpa armigera* to sugars.

**Figure supplement 1.** The inward current responses and representative traces of *Xenopus* oocytes expressing sugar GRs of *Helicoverpa armigera*.

**Figure supplement 1—source data 1.** The inward current responses of *Xenopus oocytes* expressing sugar GRs of *Helicoverpa armigera* to sugars.

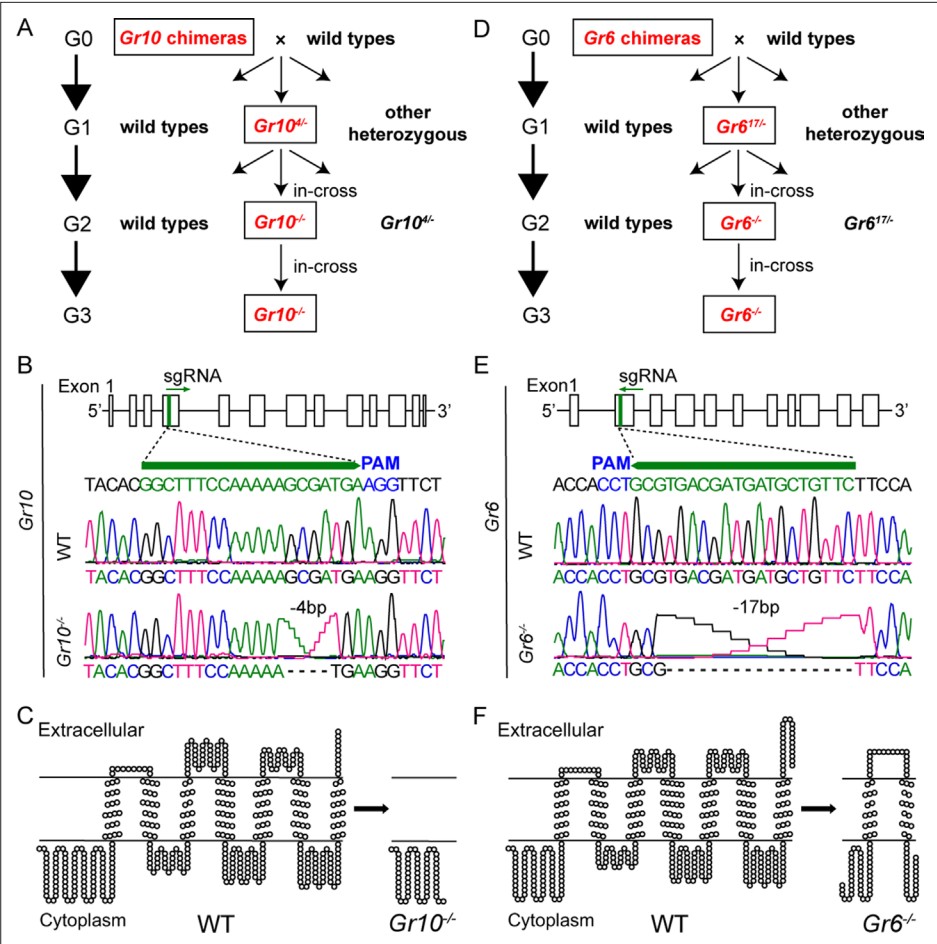

**Figure 5.** Establishment of *Gr10* and *Gr6* homozygous mutants (*Gr10*[-/-] and *Gr6*[-/-]) of *Helicoverpa armigera* via CRISPR/Cas9. (**A**) The cross process of obtaining *Gr10*[-/-]. (**B**) The genomic structure of *Gr10*, the single-guide RNA (sgRNA) targeting sequence (in green), and representative chromatograms of direct sequencing of the PCR products obtained from wild types (WT) and *Gr10*[-/-], in which 4 bp of the *Gr10* sequence were deleted. (**C**) The predicted secondary structures of the Gr10 protein in WT and the truncated Gr10 protein in *Gr10*[-/-]. (**D**) The cross process of obtaining *Gr6*[-/-]. (**E**) The genomic structure of *Gr6*, the single-guide RNA (sgRNA) targeting sequence (in green), and representative chromatograms of direct sequencing of the PCR products obtained from WT and *Gr6*[-/-], in which 17 bp of the *Gr6* sequence were deleted. (**F**) The predicted secondary structures of the Gr6 protein in WT and the truncated Gr6 protein in *Gr6*[-/-]. (**B and E**) Boxes represent exons, black lines represent introns, the green arrowhead indicates the direction of sgRNA; the protospacer adjacent motif (PAM) is in blue. (**C and F**) The secondary structure of *Gr10* and *Gr6* in WT, *Gr10*[-/-] and *Gr6*[-/-] was predicted by https://dtu.biolib.com/DeepTMHMM, and the image was constructed by TOPO2 software (http://www.sacs.ucsf.edu/TOPO2).

The online version of this article includes the following source data and figure supplement(s) for figure 5:

**Figure supplement 1.** Confirmation of the deletion of *Gr10* and *Gr6* in *Gr10*[-/-] and *Gr6*[-/-] mutants at mRNA level.

**Figure supplement 2.** The potential off-target effects detection.

**Figure supplement 2—source data 1.** The source data of potential off-target effects detection.

responsive to fucose, sucrose and fructose in the order of sensitivity, and the combination of Gr10 and Gr6 is sensitive to sucrose and then fucose, which is in line with the characteristics of the sugar GSNs in larval and adult taste organs where the two GRs are highly expressed respectively.

## Establishment of *Gr10* and *Gr6* homozygous mutants

In the above study, we discovered ligands of Gr10 and Gr6 using the ectopic expression system. To validate the function of *Gr10* and *Gr6* in vivo, we further constructed *Gr10* and *Gr6* homozygous mutants (*Gr10*[-/-] and *Gr6*[-/-], respectively) using CRISPR/Cas9 techniques (***Figure 5***). *Gr10*[-/-] and

$Gr6^{-/-}$ generated 4 bp and 17 bp deletions at the sgRNA targeting site (**Figure 5**), respectively, which also confirmed the deletion at the mRNA level in $Gr10^{-/-}$ and $Gr6^{-/-}$ (**Figure 5—figure supplement 1**). The deletion resulted in truncated Gr10 and Gr6 proteins. The Gr10 sequence was reduced from 460 to 62 amino acids and contained no transmembrane domains, whereas the Gr6 sequence was reduced from 452 to 115 amino acids and contained only two transmembrane domains (**Figure 5**).

In order to predict the potential off-target sites of sgRNA of $Gr6$ and $Gr10$, we used online software Cas-OFFinder (http://www.rgenome.net/cas-offinder/) to blast the genome of $H.\ armigera$, and the mismatch number was set to less than or equal to 3. According to the predicted results, the $Gr10$ sgRNA had no potential off-target region but $Gr6$ sgRNA had one. Therefore, we amplified and sequenced the potential off-target region of $Gr6^{-/-}$ and found there was no frameshift or premature stop codon in the region compared to WT (**Figure 5—figure supplement 2A**). It is worth mentioning that there was no potential off-target region of $Gr6$ sgRNA or $Gr10$ sgRNA in all the other eight sugar receptor genes of $H.\ armigera$. We further found there was no difference in the response to xylose of the medial sensilla styloconica among WT, $Gr10^{-/-}$ and $Gr6^{-/-}$ (**Figure 5—figure supplement 2B**). Furthermore, WT, $Gr10^{-/-}$ and $Gr6^{-/-}$ did not show differences in the larval body weight, adult lifespan, and number of eggs laid per female (**Figure 5—figure supplement 2C**). All these results suggest that no off-target effects occurred in the study.

## Effects of *Gr10* or *Gr6* knockout on sugar reception

After obtaining the two homozygous mutants, we compared electrophysiological responses of the sugar GSNs in larval lateral sensilla styloconica of the WT, $Gr10^{-/-}$, and $Gr6^{-/-}$ to sucrose and fucose using 10 mM sinigrin as a positive control, which induced the response of a bitter cell in the same sensilla. To 1 mM, 10 mM and 100 mM sucrose, the firing rates of the larval sugar GSNs in $Gr10^{-/-}$ were only 40%, 20% and 28% of those in the WT, respectively, whereas those in $Gr6^{-/-}$ larvae did not differ from those in the WT larvae (**Figure 6A–B**). To 10 mM and 100 mM fucose, the firing rates of the larval sugar GSNs in $Gr6^{-/-}$ were only 20% and 5.8% of those in the WT, respectively, whereas those of $Gr10^{-/-}$ did not differ from the WT (**Figure 6A–B**). The electrophysiological responses of the bitter GSNs to sinigrin in the larval lateral sensilla styloconica of the WT, $Gr10^{-/-}$, and $Gr6^{-/-}$ were all strong and did not differ (**Figure 6A–B**). It is clear that $Gr10$ plays a key role in sucrose reception, whereas $Gr6$ plays a key role in fucose reception of the sugar GSNs in the lateral sensilla styloconica of larvae.

Similarly, we compared the electrophysiological responses of the sugar GSNs in contact chemosensilla of the adult antennae, foreleg tarsi, and proboscis of the WT, $Gr10^{-/-}$, and $Gr6^{-/-}$ to 10 mM and 100 mM of sucrose and fucose, and 100 mM fructose. Additionally, 10 mM nicotine was used as a positive control, that induced electrophysiological responses of bitter GSNs in the top sensilla of antennae and tarsi. The firing rates of the sugar GSNs in the antennal sensilla of $Gr6^{-/-}$ to 10 mM and 100 mM sucrose, 10 mM and 100 mM fucose, and 100 mM fructose were only 32% and 9%, 22% and 11%, and 29% of those in the WT, respectively (**Figure 6C–D**). The firing rates of the sugar GSNs in antennal sensilla of $Gr10^{-/-}$ to sucrose, fucose, or fructose did not differ from those of the WT adults. The antennal sensilla of the WT, $Gr10^{-/-}$, and $Gr6^{-/-}$ all showed strong responses to nicotine with no differences (**Figure 6C–D**). The response patterns of the sugar GSNs in the foreleg tarsi of the WT, $Gr10^{-/-}$, and $Gr6^{-/-}$ to 10 mM and 100 mM sucrose, 10 mM and 100 mM fucose, and 100 mM fructose were similar to those of the sugar GSNs in antennae (**Figure 6E–F**). The firing rates of the sugar GSNs in proboscis sensilla of $Gr6^{-/-}$ to sucrose, fucose, and fructose were much lower and even close to zero (**Figure 6G–H**). The firing rates of the sugar GSNs in tarsal and proboscis sensilla of $Gr10^{-/-}$ to sucrose, fucose, and fructose did not differ from those of the WT adults (**Figure 6E–H**). The tarsal sensilla of the WT, $Gr10^{-/-}$, and $Gr6^{-/-}$ responded to nicotine with no difference (**Figure 6E–F**).

## Effects of *Gr10* and *Gr6* knockout on behavioral responses

To determine the effect of $Gr10$ and $Gr6$ knockout on feeding responses of $H.\ armigera$ larvae, we compared the larval feeding preferences of the WT, $Gr10^{-/-}$, and $Gr6^{-/-}$ to sucrose by two-choice tests. The $Gr10^{-/-}$ larvae showed no preference for the leaf discs painted with 10 mM sucrose (p=0.4258). However, they still had preference for the discs with 100 mM sucrose (p=0.0141; **Figure 7A–B**) though their PI value was decreased than those of the WT (**Figure 7B**). The $Gr6^{-/-}$ larvae were the same as the WT larvae and retained a significant preference for 10 mM or 100 mM sucrose diets (**Figure 7A–B**).

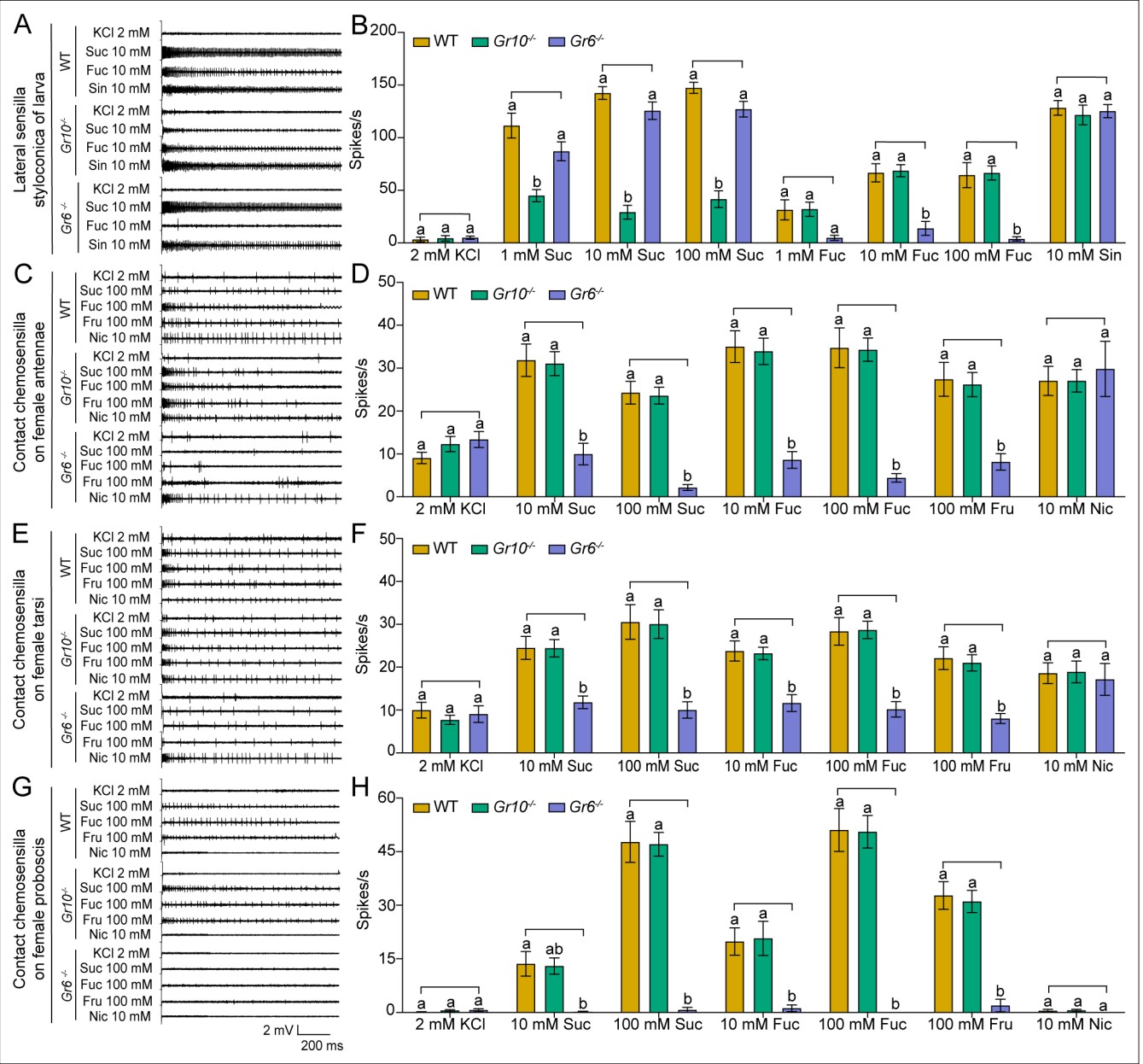

**Figure 6.** Electrophysiological responses of larval and adult contact chemosensilla in WT, *Gr10⁻/⁻* and *Gr6⁻/⁻* of *Helicoverpa armigera* to sucrose and other compounds. (**A**) The representative spike traces of lateral sensilla styloconica on larval maxillary galea. (**B**) Quantifications of the firing rates of the lateral sensilla styloconica on larval maxillary galea (mean ± SEM; WT, n=12; *Gr6⁻/⁻*, n=13; *Gr10⁻/⁻*, n=13). (**C**) The representative spike traces of contact chemosensilla on female antennae. (**D**) Quantifications of the firing rates of the contact chemosensilla on female antennae (mean ± SEM; WT, n=20. *Gr6⁻/⁻*: nicotine, n=17; other compounds, n=18. *Gr10⁻/⁻*, n=20). (**E**) The representative spike traces of contact chemosensilla on female tarsi. (**F**) Quantifications of the firing rates of the contact chemosensilla on female tarsi (mean ± SEM; WT, n=18; *Gr6⁻/⁻*: nicotine, n=14. other compounds, n=18. *Gr10⁻/⁻*, n=18). (**G**) The representative spike traces of contact chemosensilla on female proboscis. (**H**) Quantifications of the firing rates of the contact chemosensilla on female proboscis (mean ± SEM; WT: sucrose 100 mM, n=17; other compounds, n=18. *Gr6⁻/⁻*: sucrose 100 mM, n=17; other compounds, n=18. *Gr10⁻/⁻*, n=18). (**B, D, F**, and **H**) Two-way ANOVA with post hoc Tukey's multiple comparison was used separately for sucrose and fucose, and one-way ANOVA with Tukey's HSD test was used for KCl, sinigrin, fructose, and nicotine. Different letters labeled indicate significant differences ($P<0.05$). Suc: sucrose; Fuc: fucose; Sin: sinigrin; Fru: fructose; Nic: nicotine.

The online version of this article includes the following source data for figure 6:

**Source data 1.** Electrophysiological responses of larval and adult contact chemosensilla in WT, *Gr10⁻/⁻* and *Gr6⁻/⁻* of *Helicoverpa armigera* to sucrose and other compounds.

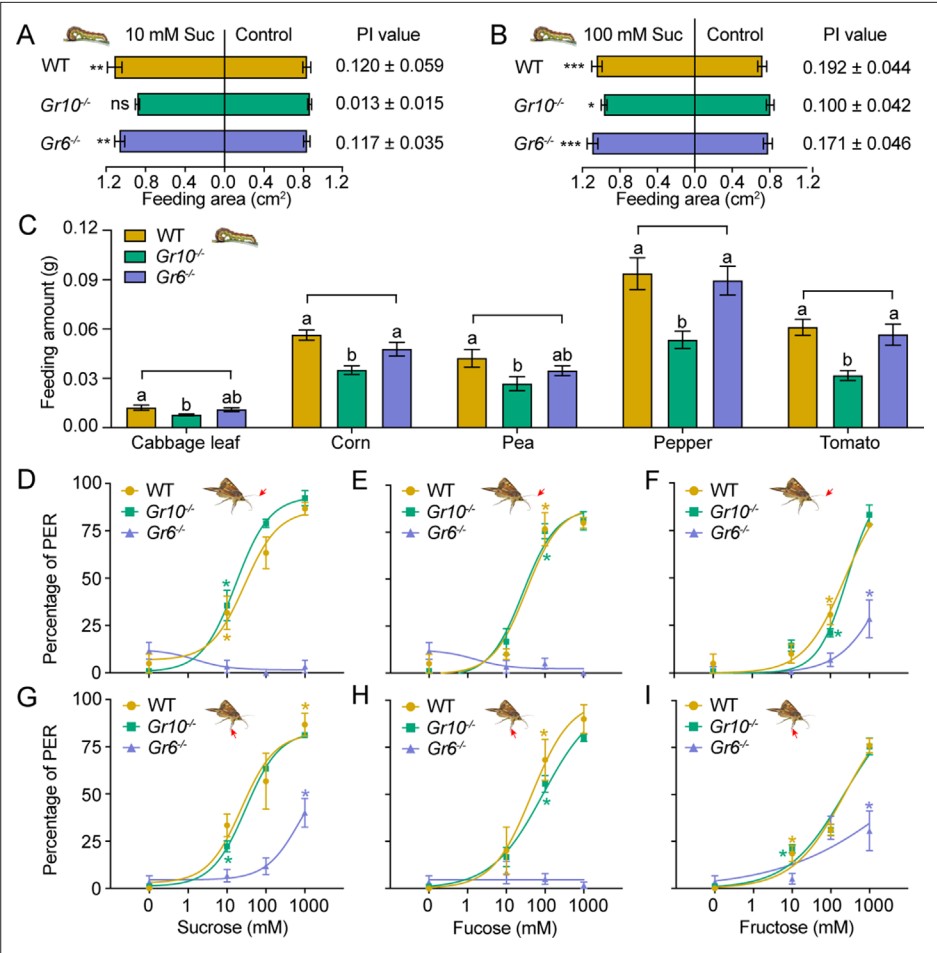

**Figure 7.** Behavioral responses of WT, *Gr10*⁻/⁻ and *Gr6*⁻/⁻ larvae and adults of *Helicoverpa armigera* to sugars and plant tissues. (**A**) Feeding area of 5th instar larvae in two-choice tests and the PI value to 10 mM sucrose (n=20). ** p<0.01; ns indicates no significance, p ≥ 0.05 (paired *t* test). Suc, sucrose. (**B**) Feeding area of 5th instar larvae in two-choice tests and the PI value to 100 mM sucrose (n=20). * p<0.05; ** p<0.01; *** p<0.001 (paired *t* test). Suc, sucrose. (**C**) Feeding amount of 5th instar larvae on cabbage leaves, corn kernels, pea seeds, pepper fruits, and tomato fruits in no-choice tests (n=20). Data were analyzed by one-way ANOVA with Tukey's HSD test, and different letters labeled on the data of WT, *Gr10*⁻/⁻ and *Gr6*⁻/⁻ for each plant tissue indicate significant differences (p<0.05). Proboscis extension reflex (PER) in adult females upon (**D**) antennal stimulation by sucrose concentrations (n=3), (**E**) antennal stimulation by fucose concentrations (n=3), (**F**) antennal stimulation by fructose concentrations (n=3), (**G**) tarsal stimulation by sucrose concentrations (n=3), (**H**) tarsal stimulation by fucose concentrations (n=3), and (**I**) tarsal stimulation by fructose concentrations (n=3). (A to I) Data are mean ± SEM. (D to I) Data were analyzed by two-way ANOVA with post hoc Tukey's multiple comparison. * p<0.05. The red arrow indicates the stimulating site.

The online version of this article includes the following source data and figure supplement(s) for figure 7:

**Source data 1.** Behavioral responses of WT, *Gr10*⁻/⁻ and *Gr6*⁻/⁻ larvae and adults of *Helicoverpa armigera* to sugars and plant tissues.

**Figure supplement 1.** Feeding responses and the PI value of 5th instar larvae of WT, *Gr10*⁻/⁻ and *Gr6*⁻/⁻ of *Helicoverpa armigera* to fructose painted on the cabbage leaf discs in two-choice tests.

**Figure supplement 1—source data 1.** Behavioral responses of WT, *Gr10*⁻/⁻ and *Gr6*⁻/⁻ larvae and adults of *Helicoverpa armigera* to fructose.

The WT, *Gr10*⁻/⁻, and *Gr6*⁻/⁻ larvae all showed similar preference for 10 mM and 100 mM fructose (*Figure 7—figure supplement 1A–B*).

We also compared the feeding amount in 1 hour of the larvae of WT, *Gr10*⁻/⁻, and *Gr6*⁻/⁻ on cabbage leaves, corn kernels, pea seeds, pepper fruits, and tomato fruits by no-choice tests. On each of these

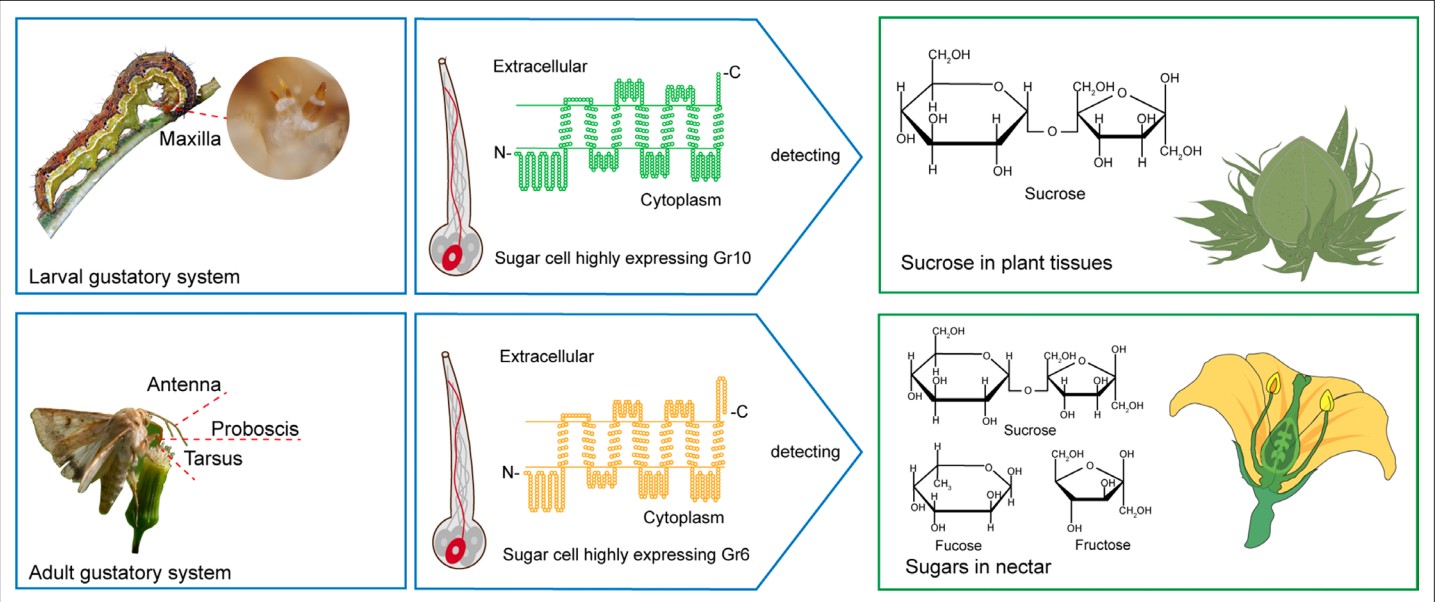

**Figure 8.** Two gustatory receptors in the cotton bollworm, *Helicoverpa armigera* mainly mediate taste sensation of sugars in the larval and adult foods.

plant tissues, the *Gr10⁻/⁻* larvae fed significantly less than the WT larvae, but the *Gr6⁻/⁻* larvae fed the same amount as the WT larvae (*Figure 7C*).

Moreover, we examined the PER of the WT, *Gr10⁻/⁻*, and *Gr6⁻/⁻* female adult antennae and tarsi in response to stimulation of sucrose, fucose, and fructose at different concentrations. When antennae or tarsi were stimulated with 10 mM, 100 mM, or 1000 mM sucrose, fucose, or fructose, *Gr6⁻/⁻* females showed a considerably lower PER percentage than the WT females and even had little or no PER in response to sucrose and fucose, whereas the PER percentage of *Gr10⁻/⁻* females was not different from that of the WT females (*Figure 7D–I*).

In conclusion, *Gr10* plays a key role in sucrose reception by the sugar GSNs in the larval lateral sensilla styloconica, and mediates the feeding preference of larvae to sucrose, whereas *Gr6* plays a key role in sensing sucrose, fucose, and fructose by the sugar GSNs in contact chemosensilla of the adult antennae, foreleg tarsi and proboscis, and mediates the related PER behavior of adults.

## Discussion

In this study, we reveal the different molecular bases of sucrose reception between *H. armigera* larvae and adults through ectopic expression and genetic editing of *GR*s. Gr10, which was highly expressed in the larval maxillary galea, is specifically tuned to sucrose with high sensitivity, whereas Gr6, which was highly expressed in the adult antennae, foreleg tarsi and proboscis, has a broader response spectrum but lower sensitivity to sucrose, indicating that larvae mainly use Gr10 to detect low sucrose in plant tissues, while adults primarily use Gr6 to detect a variety of sugars with high content, including sucrose in nectar (*Figure 8*).

Our findings indicate that the *H. armigera* larvae and adults adopt different sugar-sensing systems to satisfy their requirements for food selection. The larvae mainly feed on young leaves, flower buds, and fruits of host plants. As a universal phagostimulant, sucrose is the major sugar in plants and considerably varies in content in different tissues and developmental stages. Tomato is a principal host plant of *H. armigera*. The sucrose content in its leaves generally ranges from 13.72 to 42.06 mM, but that in its fruits generally ranges from 35.2 to 988.22 mM (*Schauer et al., 2005*). The adults mainly feed on plant nectar. Sucrose and some monosaccharides, especially fructose and glucose, are the main components in nectar, where the sugar content is much higher than that in leaves and fruits. The sugar content in the nectar of cotton plants is 10–34% (which roughly corresponds to 0.5 M–1.7 M; *Vansell, 1944*; *Waller et al., 1981*), and that of soybean plants is even higher, around 23–60% (which roughly corresponds to 1.2–3 M; *Free, 1993*). It is clear that, in their life cycle, a single sugar-sensing

mechanism cannot meet the requirements for food selection by this phytophagous insect species. From the perspective of insect and plant co-evolution, using value-based and target-specific sugar-sensing systems is a favorable choice for phytophagous insects to ensure their growth and reproduction on plants.

Although Gr10 plays a major role in sucrose reception in the larval stage, the response of the sugar GSNs in larval lateral sensilla styloconica to sucrose did not completely disappear in $Gr10^{-/-}$ (*Figure 6A–B*), and the larvae still showed a behavioral preference for the high concentration of sucrose (100 mM; *Figure 7B*). It is most likely that low-level expressed Gr6 is functional in the GSNs. The PER percentage of $Gr6^{-/-}$ adults was not disappeared when the antennae and tarsi were stimulated by high fructose concentrations (*Figure 7F and I*); this indicates that fructose-tuned GRs other than *Gr6* were involved. Previous studies have shown that *Gr9* (*Gr4* in *Jiang et al., 2015*), as an ortholog of DmelGr43a and BmorGr9, is a fructose-tuned GR (*Ai et al., 2022*; *Jiang et al., 2015*; *Sato et al., 2011*; *Xu et al., 2012*). From our results, knocking out *Gr10* or *Gr6* is unlikely to be compensated by overexpression of other sugar GRs. One of our recent studies showed that *Orco* knockout had no significant effect on the expression of most OR, IR and GR genes in adult antennae of *H. armigera* although some genes were up- or down-regulated to a certain extent (*Fan et al., 2022*). To elucidate the precise molecular mechanisms of sugar reception in *H. armigera* is necessary to compare a series of single, double and even multiple Gr knock-out lines and investigate the downstream effects of the GRs.

The larval maxillary galea with only two sensilla styloconica is an ideal system for studying the mechanisms of sugar sensing. In each maxillary galea of *H. armigera*, the only sugar GSN sensitive to sucrose and fucose is located in the lateral sensillum styloconicum. Based on our results, we speculate that Gr10 and Gr6 are expressed in this GSN with different levels, and mainly determine the response profile of the cell. The firing rate of the cell in $Gr10^{-/-}$ to 1 mM, 10 mM, and 100 mM sucrose was significantly lower than that in WT, but there was no difference between WT and $Gr6^{-/-}$, indicating that Gr10 mainly determine the sensitivity of this cell to sucrose (*Figure 6A–B*). In contrast, the firing rate of the cell in $Gr6^{-/-}$ to 10 mM and 100 mM fucose was significantly lower than that in the WT, but there was no difference between WT and $Gr10^{-/-}$, indicating that Gr6 was also functional in larvae (*Figure 6A–B*). This can be well explained by the response profiles of the oocytes expressing Gr10, Gr6, and their combination (*Figure 4*).

Sugars are common phagostimulants in animals, but the molecular basis of sugar perception differs among animals. In mammals, the taste receptors T1R2 and T1R3 combine to complete the perception of all carbohydrate compounds, including sucrose, and both receptors are indispensable (*Damak et al., 2003*; *Li et al., 2002*; *Nelson et al., 2001*; *Zhao et al., 2003*). In *D. melanogaster* adults, multiple sugar GRs are involved in sugar reception, different GRs expressed in sweet taste neurons may function as multimers (*Dahanukar et al., 2001*; *Dahanukar et al., 2007*; *Fujii et al., 2015*; *Jiao et al., 2007*; *Jiao et al., 2008*; *Slone et al., 2007*). Gr5a and Gr64a are widely used for responses to complementary subsets of sugars in labellar sweet taste neurons (*Dahanukar et al., 2007*), and Gr64f is also required broadly as a coreceptor for the detection of sugars (*Jiao et al., 2008*). The recent successful ectopic expression of single sugar GRs in olfactory neurons confirmed that each sugar GR is directly involved in ligand recognition (*Freeman et al., 2014*). Sweet sensing neurons or receptors have not been identified in the larval peripheral taste system (*Kwon et al., 2011*). Sugar perception in *Drosophila* larvae has been assigned to the fructose receptor *Gr43a*, which, however, is expressed in pharyngeal sensory organs, foregut, and in the brain, but absent from external taste neurons (*Mishra et al., 2013*). This is most likely due to the fact that fruit fly larvae prefer to feed on rotting fruits at the optimum stage of fermentation, which contain fructose as the main sugar. Anyway, the sucrose sensing neurons were recently identified in the larval primary taste center (*Maier et al., 2021*). In honeybees, AmGr1 acts as a GR for certain sugars other than fructose, and AmGr2 acts as a coreceptor with AmGr1 (*Jung et al., 2015*). AmGr3 is tuned to fructose and is phylogenetically closely related to the DmGr43a (*Değirmenci et al., 2020*; *Takada et al., 2018*). The whitefly *Bemisia tabaci* consistently choose diets containing higher sucrose concentrations. It has been shown that BtabGr1 displayed significant sucrose specificity when expressed in *Xenopus* oocytes, and silencing of this GR significantly interfered with the ability of *B. tabaci* adults to detect sucrose in the phloem sap (*Aidlin Harari et al., 2023*).

The most studied phytophagous insects for sugar perception are Lepidopterans. BmGr9 in *B. mori* (*Morinaga et al., 2022*; *Sato et al., 2011*), HarmGr9 in *H. armigera* named as HarmGr4 by *Jiang et al., 2015*; *Xu et al., 2012*, PxGr43a-1 in *P. xylostella* (*Liu et al., 2020*), and SlGr8 in *Spodoptera litura* (*Liu et al., 2019*) were all the orthologs of DmGr43a in *D. melanogaster* (*Miyamoto et al., 2012*), and tuned to fructose without complexing with other GRs. BmGr4 was recently identified as a receptor for sucrose and glucose and may function as a chemosensor for nutritional compounds in both midgut and brain (*Mang et al., 2022*). In this study, we find *Gr10* and *Gr6* were highly expressed in the larval and adult taste organs, respectively. The experiments on ectopic expression in *Xenopus* oocytes shows that Gr10 specifically responded to sucrose, Gr6 responded to fucose and sucrose, and weakly to fructose, and the combination of Gr10 and Gr6 responded to sucrose and weakly to fucose (*Figure 4A–B*). These results indicate that Gr10 and Gr6 are capable of forming functional receptors singly, and the function of the two co-expressed receptors is not equal to the sum of the functions of the two receptors (*Figure 4*). When further tracing the genomic locations of the two GR genes, we confirm that *Gr10* and *Gr6* together with other 6 sugar GR genes (except for *Gr9*) are tandemly duplicated on the same scaffold of *H. armigera* (*Pearce et al., 2017*). The sugar GR gene cluster and interactions between GRs could provide a degree of evolutionary and regulatory flexibility that may meet the needs of the insect to sense subtle sugar components and concentrations in food sources in a changing environment (*Delventhal and Carlson, 2016*). In this study, the expression patterns of nine sugar GRs in three taste organs of adult *H. armigera* show that there is a disparity in GRs, specifically GR5 and GR6, between the female antenna, tarsi and proboscis, which may be an evolutionary adaptation reflecting subtle differentiation in the function of these taste organs in adult foraging. Antennae and tarsi play a role in the exploration of potential sugar sources, while the proboscis plays a more precise role in the final decision to feed.

In this work, we report for the first time the molecular basis of sucrose reception in external taste neurons of a Lepidopteran insect, and discover that different taste receptors are used for food selection in *H. armigera* larval and adult stages. These findings greatly improve our understanding of "sweet" taste of phytophagous insects. According to the phylogenetic tree of sugar GRs in insects, *Gr10* is clustered in the same branch with BmorGr5 of *Bombyx mori*, whereas *Gr6* is clustered in another branch with BmorGr6 (*Figure 3A*). GRs closely associated with *Gr10* and *Gr6* are also found in other Lepidopteran species such as *D. plexippus*, *H. melpomene*, and *P. xylostella* (*Figure 3A*). Therefore, we speculate that similar sugar-sensing mechanisms may also exist in other Lepidopteran species, which is worth verifying in these species in the future. Knockout of *Gr10* or *Gr6* led to a significant decrease in sugar sensitivity and food preference of the larvae and adults of *H. armigera*, respectively, which is bound to bring adverse consequences to the survival and reproduction of the insects. Therefore, studying the molecular mechanisms underlying sugar perception in phytophagous insects would provide new insights into the behavioral ecology of this highly diverse and important group of insects, and may also provide new means for agricultural pest control by blocking or disrupting sugar receptors.

# Materials and methods

## Insects

*Helicoverpa armigera* were collected from Xuchang (Henan Province, China) and raised in an indoor environment (27 ± 1°C, 70–80% humidity) with a 16 : 8 hr light: dark cycle. The larvae fed on artificial diet (mainly including wheat bran, wheat germ, and tomato paste) in a glass tube (I.D., 2.5 cm; length, 8 cm). When the larvae reached the 3rd instar, they were transferred to a new glass tube and raised individually until pupation. Female and male moths were raised separately until they became sexually mature and then moved to a cage for mating. Since the taste system of the female is similar to that of the male and the nutrient requirements are more critical for females (*Jiang et al., 2015*; *Zhang et al., 2010*), the virgin female adults were used in the related experiments.

## *Xenopus laevis*

*Xenopus laevis* were fostered at 20 ± 1°C in the Animal Laboratory Center, Institute of Genetics and Developmental Biology, Chinese Academy of Sciences. They fed on artificial frog food (mainly including fish meal, yeast extract powder, high-gluten flour). All experiments were approved by the

Animal Care and Use Committee of the Institute of Zoology, Chinese Academy of Sciences and followed the Guide for the Care and Use of Laboratory Animals (IOZ17090-A).

## Scanning electron microscopy

Just as our previous study on tarsal contact chemosensilla on fore legs of female moths (*Zhang et al., 2010*), the samples of antennae or proboscis in adult female *H. armigera* were mounted directly on stainless steel sample buds and sputter-coated with a 10-nm-thick layer of gold. Photomicrographs were obtained with a scanning electron microscope (HITACHI SU8010) in the Institute of Microbiology, Chinese Academy of Sciences.

## Tip-recording

The tip-recording technique based on a previously described methods (*Hodgson et al., 1955*; *van Loon, 1990*; *Zhang et al., 2010*) was used to record the electrophysiological responses of the lateral and medial sensilla styloconica on larval maxillary galea and the contact chemosensilla on the top areas of adult antennae, tarsi and proboscis of *H. armigera*. A glass capillary filled with the test compound solution, into which a silver wire was inserted, was placed in contact with the contact chemosensillum to be recorded as the recording electrode. The tip diameter of the electrode was about 50 μm, suitable for recording from a single sensillum. Action potentials (spikes) generated during the first second after stimulus onset were amplified by a preamplifier and was sampled with a computer equipped with a Metrabyte DAS16 A/D conversion board. The amplifier used an AD 515 K (Analog Devices) integrated circuit in the first stage, yielding <1 pA input bias current, 1015 Ohm and 0.8 pF input impedance. An interface (GO-box) was used for signal conditioning. This involved a second-order band pass filter (–3 dB frequencies: 180 and 1700 Hz). Stimuli were dissolved in 2 mM KCl, which was used as the control. Two pentoses (arabinose and xylose), four hexoses (fructose, fucose, galactose, and glucose), and two disaccharides (sucrose and trehalose) (Appendix 1—key resources table) were used as sugar stimuli. Each sugar first tested at 10 mM to screen active compounds, which were then tested in a range of concentrations in dose-response assays (0.01 mM, 0.1 mM, 1 mM, 10 mM, and 100 mM). To avoid possible adaptation of the contact chemosensilla tested, the interval between two successive stimulations was at least 3 min. Digitized traces were analyzed by means of SAPID Tools software version 3.5 (*Smith et al., 1990*) and Autospike 3.7 (Syntech, Buchenbach, Germany). The spikes were sorted on the basis of typical biphasic waveforms and spike amplitudes, and the frequency of the same type of spikes was counted from the first second after stimulation.

In recording of electrophysiological responses of larvae, the first day of larvae in the 5th instar were used. To guarantee all test caterpillars were in the same stage, we first selected 4th instar larvae which were preparing to undergo ecdysis, and then transferred them to a glass tube with green pepper. The test larva was starved for approximately 2 hr before recording. The head was cut and fixed in a silver holder which was connected to a preamplifier with a copper miniconnector. The lateral and medial sensilla styloconica on larval maxillary galea were exposed for testing electrophysiological responses. About 12–13 replicates were set for each treatment.

The virgin female moths on the second day after eclosion were used for recording electrophysiological responses (*Jiang et al., 2015*; *Zhang et al., 2010*). They were only fed with ddH$_2$O. When testing sensilla on a tarsus, the foreleg was cut between coxa and femur, and the silver holder connected to the amplifier was inserted into the femur. When testing sensilla on the antennae and proboscis, the head and thorax without all legs were reserved. The silver holder was inserted into the thorax. The orientation of the taste organ was gently adjusted to allow the sensilla to be touched by the electrode. To guarantee the sensilla was steadily exposed, the distal part of antenna or proboscis was fixed on a small platform with double sided adhesive tape. For each individual, three sensilla on the top areas of antennae, foreleg tarsi, or proboscis were chosen for testing electrophysiological responses, and about 15–21 replicates were used for each treatment.

## Two-choice tests

We determined the behavioral responses to sugars of the *H. armigera* larvae using the two-choice test as previously described with small modifications (*Chen et al., 2022*). We used the leaf of cabbage (*Brassica oleraceaas*, Shenglv-7) as the substrate since it is not a favorite food for larvae. Circular cabbage leaf discs of 1 cm in diameter were prepared. The upper surface of the treated leaf disc

was painted with the 35.2 μL sugar solution with 0.01% Tween 80 (CAS: 9005-55-6, Sigma-Aldrich, St Louis, USA) using a paintbrush, and the control leaf disc was painted with 35.2 μL ddH$_2$O with 0.01% Tween 80. The 0.01% Tween 80 was added to reduce the surface tension of the cabbage leaves and increase the malleability of the test sugar solution (*Blaney and Simmonds, 1988*). The feeding preference of larvae to eight sugars including arabinose, fructose, fucose, galactose, glucose, sucrose, trehalose, and xylose (Appendix 1—key resources table) was tested at 10 mM. 0.01 mM, 0.1 mM, 1 mM, 10 mM, and 100 mM sucrose and fructose were used in a dose-response assay. The intact and the remaining leaf discs were scanned by DR-F120 scanner (Canon, Tokyo, Japan), and the area of intact and remaining leaf discs were calculated by ImageJ softwere (NIH) (*Abràmoff et al., 2004*; *Chen et al., 2022*). The feeding area = the area of intact leaf disc – the area of remaining leaf disc. The preference index (PI) was calculated according to the following formula: PI = (the feeding area of test leaf discs – the feeding area of control leaf discs) / (the feeding area of test leaf discs +the feeding area of control leaf discs). Twenty replicates were run.

## Adult proboscis extension reflex (PER) tests

We determined the behavioral responses to sugars of the *H. armigera* female adults using the proboscis extension reflex (PER) test as previously described with some modifications (*Jiang et al., 2015*). Arabinose, fructose, fucose, galactose, glucose, sucrose, trehalose, and xylose (Appendix 1—key resources table) were first tested at 100 mM to screen active compounds, which were then tested in a range of concentrations in dose-response assays (0.01 mM, 0.1 mM, 1 mM, 10 mM, 100 mM, and 1000 mM). In brief, the virgin females on the first day after eclosion were used. Each moth was harnessed in a 1 mL pipette with a tip removed, from which its head protruded, and its antennae, proboscis and forelegs were permitted to move freely. They were allowed to adapt to the environment for about 12 hr, and satiated with ddH$_2$O until the trial. The testing moth was stimulated to the distal part of one antenna with a given test solution in an absorbent cotton ball, and the PER in response to the stimulation was recorded. The interval between two successive stimulations was 5 min, during which the distal part of the antenna was washed, and the insect was fed with ddH$_2$O if needed. Four hours later, two foreleg tarsi of the moth were simultaneously stimulated with the test solutions in the same way as the antenna was stimulated, and the PER was recorded. 20–30 individual moths were used for each set of tests. The experiments were repeated three times with groups of insects from three different rearing batches.

## RNA extraction and cDNA synthesis

The galea of larvae, and the antennae, foreleg tarsi, proboscis, and abdomen of both male and female adults were removed and directly placed into 1.5 mL Eppendorf tubes (Axygen, Tewksbury, MA, USA) containing 1 mL QIAzol Lysis Reagent (QIAGEN, Hilden, Germany); then, the extraction of total RNA was carried out following the manufacturer's protocols. After extraction, the RNA quality and concentration were determined with a NanoDrop 2000 spectrophotometer (Thermo Fisher Scientific, Waltham, MA, USA). First, for cDNA synthesis 2 μg total RNA was mixed with 1 μL oligo-dT (Promega, Madison, WI, USA); then, the mixture was heated at 70 °C 5 min and followed by an ice bath for 5 min. M-MLV reverse transcriptase (Promega, Madison, WI, USA) was added; subsequently, the mixture was incubated at 42 °C for 90 min. The products were dissolved in ddH$_2$O and stored at −20 °C for later use.

## Gene cloning of sugar GRs

According to the reported sugar GR sequence of *H. armigera* (the GenBank accession number is provided in *Supplementary file 1*) for PCR cloning (*Pearce et al., 2017*; *Xu et al., 2017*), we designed specific primers (*Supplementary file 2*). The GenBank accession number of *Gr9* (named as *Gr4* in *Jiang et al., 2015*) nucleic acid sequence and protein sequence were JX982536.2 and AKG90011 in *Jiang et al., 2015*, respectively (*Jiang et al., 2015*). The Q5 High-fidelity DNA Polymerase (New England Biolabs, Beverly, MA, USA) was used to clone sugar sequences except *Gr10*. The PCR instrument was Applied Biosystems (Thermo Fisher Scientific, Woodlands, Singapore) and the sequence amplification procedure was as follows: 98 °C for 10 s, 40 cycles (60 °C for 30 s, 72 °C for 1 min), and final extension at 72 °C for 2 min. TransStart FastPfu DNA Polymerase (TransGen Biotech, Beijing, China) was used to clone the *Gr10* sequence. The amplification procedure was as follows: 95 °C for

1 min, 40 cycles (95 °C for 20 s, 50 °C for 20 s, 72 °C for 45 s), and 72 °C for 5 min. PCR products were purified, subcloned into pGEM-T vector according to the manufacturer's instructions (Promega), and sequenced. *Gr4* (402 amino acids) and *Gr7* (428 amino acids) were obtained from the tissues of male and female *H. armigera* tarsi, respectively. *Gr5* (482 amino acids), *Gr6* (452 amino acids), and *Gr10* (460 amino acids) were all obtained from the tissues of female *H. armigera* proboscis. *Gr8* (427 amino acids) and *Gr9* (474 amino acids) was obtained from the tissues of female *H. armigera* antennae. Finally, *Gr11* (435 amino acids) and *Gr12* (434 amino acids) were obtained from the tissues of female and male *H. armigera* abdomens, respectively. No alternatively spliced transcript variants were found in the gene cloning of all the sugar GRs studied.

*Xu et al., 2017* reported that *Gr10* was a pseudogene because the nucleotide sequence changed at positions 521–524 compared with the sequence reported in *Pearce et al., 2017*, and it terminated early during translation. The *Gr10* sequence that we obtained had no termination on the same position. Aligning the *Gr10* sequences of *Pearce et al., 2017* and *Xu et al., 2017* revealed that *Gr10* in *Xu et al., 2017* had 22 more amino acids at the C-terminus than that in *Pearce et al., 2017*. Therefore, we designed upstream and downstream primers of *Gr10* from the N-terminus in *Pearce et al., 2017* and the C-terminus in *Xu et al., 2017*. Finally, we obtained the full-length sequence of *Gr10* with 460 amino acids.

## Phylogenetic analysis of sugar GRs

The phylogenetic tree was built with *H. armigera* and sugar GRs of eight other insect species: Diptera, *Drosophila melanogaster* (*Robertson et al., 2003*), *Aedes aegypti* (*Kent et al., 2008*), and *Anopheles gambiae* (*Hill et al., 2002*); Hymenoptera, *Apis mellifera* (*Robertson and Wanner, 2006*); and Lepidoptera, *Danaus plexippus* (*Zhan et al., 2011*), *Heliconius melpomene* (*Briscoe et al., 2013*), *Bombyx mori* (*Guo et al., 2017*), and *Plutella xylostella* (*Engsontia et al., 2014*). Amino acid sequences were aligned with MAFFT version 7.455 (*Rozewicki et al., 2019*), and gap sites were removed with trimAI version 1.4 (*Capella-Gutiérrez et al., 2009*). The phylogenetic tree was constructed using the maximum likelihood method in IQ-tree version 6.8 (*Nguyen et al., 2015*). The bootstrap algorithm was based on the Jones–Taylor–Thornton (JTT)+F + G4 model and run 5000 times. Visualization and later modification of phylogenetic trees were performed in FigTree version 1.4.4 (http://tree.bio.ed.ac.uk/software/figtree/).

## Quantitative real-time PCR

The relative expression levels of sugar GRs in *H. armigera* were analyzed by quantitative real-time PCR (qRT-PCR) (*Livak and Schmittgen, 2001*). The specific primers (*Supplementary file 3*) were designed by Primer3plus (http://www.primer3plus.com/cgi-bin/dev/primer3plus.cgi). qRT-PCR was conducted with a QuantStudio 3 Real-Time PCR System (Thermo Fisher Scientific, Waltham, MA, USA). All reactions included 10 µL SYBR Premix Ex *Taq* (TaKaRa, Shiga, Japan); 0.4 µL (10 mM) primer (forward and reverse), 2 µL template DNA, 0.4 µL ROX Reference Dye II Dye, 6.8 µL ddH$_2$O (20 µL total). The procedure was as follows: 95 °C for 30 s; 40 cycles of 95 °C for 5 s and 60 °C for 34 s; 95 °C for 15 s; 60 °C for 1 min; and 95 °C for 15 s. *18* S ribosomal RNA (GenBank number: KT343378.1) was used as the standard gene because the Ct value of this gene was not different between different tissues (*Sharath Chandra et al., 2014*). The relative expression levels of the specific genes were quantified by using the $2^{-\Delta Ct}$ method, where ΔCt is the Ct value of GR genes subtracted from that of *18* S. Three biological replicates were run for all experiments, and there were 30–100 insects in each replicate.

## Ectopic expression and functional analysis of sugar GRs

The sugar GRs sequences which were used for ectopic expression and functional identification need to contain restriction enzyme sites and Kozak sequences (the primer sequences of *H. armigera* sugar GRs for functional identification are provided in *Supplementary file 4*). The full-length sequence of the sugar receptor was cloned from cDNA and transferred into pGEM-T vector (Promega, Madison, WI, USA). After cleavage by appropriate endonucleases, it was linked to pCS2$^+$ vector. pCS2$^+$ vector was linearized and cRNA was synthesized according to the protocol of the mMESSAGE mMACHINE SP6 Kit (Ambion, Austin, TX, USA). The products were dissolved in ddH$_2$O, diluted to 3000 ng/µL, and then stored at −80 °C until use.

Each of Gr4, Gr5, Gr6, Gr7, Gr8, Gr10, Gr11, Gr12, Gr5 +Gr6, and Gr10 +Gr6 was expressed in oocytes. The acquisition of *X. laevis* oocytes was performed following a previously described protocol (***Nakagawa and Touhara, 2013***). *X. laevis* were anesthetized by a mixture of ice and water for 30 min; then, the abdomen was cut to obtain oocytes. The *Xenopus* oocytes were treated with 2 mg/mL collagenase type I (Sigma-Aldrich, St Louis, MO, USA) in a calcium-free saline buffer (82.5 mM NaCl, 2 mM KCl, 1 mM MgCl$_2$, 5 mM HEPES, pH = 7.5; the reagents are provided in Appendix 1—key resource table) at room temperature for 25 min. The oocytes were microinjected with 27.6 nL (80 ng) cRNA of each *GR* by SMART Touch (World Precision Instruments, Sarasota, FL, USA). For Gr10 +Gr6 or Gr5 +Gr6, a cRNA mixture with a ratio of 1:1 was used. The oocytes microinjected with RNAse-free water were used as the control. Then, the oocytes were incubated for 3–4 days at 16 °C in the bath solution (96 mM NaCl, 2 mM KCl, 1 mM MgCl$_2$, 1.8 mM CaCl$_2$, 5 mM HEPES, pH 7.5; reagents are provided in Appendix 1—key resource table).

The whole-cell current was recorded with a two-electrode voltage clamp, the procedures as previously described (***Jiang et al., 2015***). The glass electrode, filled with 3 M KCl, was inserted into the oocytes and maintained at 0.2–2.0 MΩ. The signal was amplified with an OC-725C amplifier (Warner Instruments, Hamden, CT, USA) at a holding potential at –80 mV all the time, low-pass filtered at 50 Hz, and digitized at 1 kHz. The sugars (Appendix 1—key resource table) were dissolved in 1×Ringer solution (96 mM NaCl, 2 mM KCl, 1 mM MgCl$_2$, 1.8 mM CaCl$_2$, 5 mM HEPES, pH 7.5) as the test solution. The oocytes were stimulated by test solution via the perfusion system for 20 s each time, and then washed by 1×Ringer solution (***Nakagawa and Touhara, 2013***). The concentration of 100 mM for each chemical was randomly used at first, and then concentration gradients (ranging from 0.01 mM, 0.1 mM, 1 mM, 10 mM, 50 mM, 100 mM, 250 mM, and 500 mM) were recorded later when a clear current response was detected. Data were obtained and analyzed using Digidata 1322 A and pCLAMP software version 10.4.2.0 (Axon Instruments Inc, Foster City, CA, USA).

## In vitro synthesis of single-guide RNA

*Gr10* and *Gr6* single-guide RNA (sgRNA) position screening was conducted at http://www.rgenome.net/cas-designer/. The sgRNA target site of *Gr10* was designed on exon 4 (***Figure 5B***), and the sgRNA target site of *Gr6* was designed on exon 2 (***Figure 5E***). The *Gr10* and *Gr6* sgRNA primers (***Supplementary file 5***) were designed based on the principles provided by the GeneArt Precision gRNA Synthesis Kit manual (Invitrogen, Vilnius, Lithuania). The sgRNAs were synthesized by the GeneArt gRNA Prep Kit (Invitrogen, Vilnius, Lithuania), and purified by the GeneArt gRNA Clean Up Kit (Invitrogen, Vilnius, Lithuania). The products were dissolved in RNAse-free water and then stored at −80 °C until use. TrueCut Cas9 protein 2 (Invitrogen, Vilnius, Lithuania) was diluted to 500 ng/mL and stored at −20 °C.

## Embryo microinjection

The fresh laid eggs of *H. armigera* were placed on microscope slides and fixed on double-sided adhesive tape. These operations were finished within 2 hr after oviposition. Microinjection into these pretreated eggs was performed with a mixture of *Gr10* (or *Gr6*) sgRNA and Cas9 protein, with final concentrations at 250 ng/μL and 100 ng/μL, respectively. Microinjection was finished using a PLI-100A PICO-INJECTOR (Warner Instruments, Hamden, Connecticut, USA). The injected eggs were placed in a 10 cm Petri dish, covered with wet filter paper, and incubated at 25 °C until hatching.

## DNA extraction and mutagenesis detection

To screen *Gr10* (or *Gr6*) homozygous mutants, the tarsi on hindlegs were used for DNA extraction with the Animal Tissue PCR Kit (TransGen Biotech, Beijing, China). Then, we used PCR to amplify the sequences that covered *Gr10* or *Gr6* sgRNA targeting sites. The reactions were 50 μL total and included: 2 μL template, 1 μL forward primer, 1 μL reverse primer (mutation detection primer sequences are provided in ***Supplementary file 5***), 25 μL Premix *Taq* (Takara, Dalian, China), and 19 μL ddH$_2$O. The procedure was as follows: 94 °C for 3 min, 40 cycles [94 °C for 30 s, 55 °C (*Gr6*), or 60 °C (*Gr10*) for 30 s], 72 °C for 1 min, and finally 72 °C for 5 min. The PCR products were sent to Beijing Genomics Institute (BGI) or the SinoGenoMax for Sanger sequencing, and the sequence results were analyzed by SnapGene software version 4.3.8 (from Insightful Science; available at https://www.snapgene.com/) and SeqMan software version 7.1 (DNASTAR, Madison, WI, USA).

## Screening homozygous mutants

*Gr10* and *Gr6* homozygous mutant lines (*Gr10*$^{-/-}$ and *Gr6*$^{-/-}$, respectively) were established. A mixture of Gr10 sgRNA and Cas9 protein was co-injected into each of 732 eggs that were laid within 2 hr. After hatching, the larvae were raised on artificial diet, and finally 125 G0 adults were obtained. Among 125 G0 adults, 15 G0 harbored multiple targeted mutations. Each mutated G0 was backcrossed with the wild type to obtain G1 offspring. From 57 G1 adults, one DNA strands containing 4 bp-deleted modifications, were in-crossed to establish stable lines; for the expanding mutant population, 4 bp-deleted homozygotes of G2 moths were in-crossed to generate homozygotes G3 moths (*Figure 5A*). Similarly, a mixture of Gr6 sgRNA and Cas9 protein was co-injected into each of 695 eggs. After hatching, the larvae were raised on artificial diet, and finally 132 G0 adults were obtained. Among 132 G0 adults, 12 G0 harbored multiple targeted mutations. From 66 G1 adults, one DNA strands containing 17 bp-deleted modifications, were in-crossed to establish stable lines; for the expanding mutant population, 17 bp-deleted homozygotes of G2 moths were in-crossed to generate homozygotes G3 moths (*Figure 5D*). The related mutation rates are provided in *Supplementary file 6*.

To confirm the deletion in *Gr10*$^{-/-}$ and *Gr6*$^{-/-}$ at mRNA level, the total RNA of the female adult proboscis of WT, *Gr10*$^{-/-}$ and *Gr6*$^{-/-}$ were extracted, and the cDNA was synthesized following the previous gene cloning protocol. The specific primers (*Supplementary file 5*) were designed based on the DNA exon sequences of *Gr10* and *Gr6*, which covered the DNA-confirmed sgRNA targeting sites. During mRNA sequence cloning, the PCR reaction conditions are largely the same as those used for DNA sequence confirmation, with little modification (the annealing temperature reduced to 50 °C). The nucleic acid sequences of the identified *Gr10* and *Gr6* were aligned manually by tools at the T-Coffee web server (*Notredame et al., 2000*).

## Off-target effects detection

The potential off-target sites were predicted as previously described (*Chen et al., 2020*). The online software Cas-OFFinder (http://www.rgenome.net/cas-offinder/) was used to blast the *Gr10* and *Gr6* sgRNAs against the *H. armigera* genome (*Pearce et al., 2017*), with mismatch number ≤3. According to the blast results, *Gr10* has no potential off-target site, and one sequence (GenBank accession number: LOC126054697) in the presence of *Gr6* sgRNA showed a potential off-target region. The specific primers (*Supplementary file 5*) were designed based on the DNA sequence of LOC126054697 covering the potential off-target sites. The DNA extraction and mutagenesis detection methods were described previously. Five individual replicates were performed.

To further illustrate that the mutation strains did not exhibit off-target effects, the electrophysiological responses of larvae and the biological parameters were compared among WT, *Gr10*$^{-/-}$, and *Gr6*$^{-/-}$. The larval medial sensilla styloconica to xylose, the body weight of the fifth instar larvae on the first day, the adult lifespan, and the number of eggs laid were recorded. Five to sixteen individuals were recorded.

## Effects of *Gr10* or *Gr6* knockout on sugar reception in larvae and adults

We recorded the electrophysiological responses of contact chemosensilla in the wild type (WT), *Gr10*$^{-/-}$, and *Gr6*$^{-/-}$ by the tip-recording technique as described earlier. Concentration of 1 mM, 10 mM, and 100 mM for sucrose and fucose were used as stimuli for lateral sensilla styloconica in larvae. As the response thresholds of sugar GSNs in adults were higher than those in larvae, concentrations of 10 mM and 100 mM for sucrose and fucose and 100 mM for fructose were used for adult contact chemosensilla. 2 mM KCl was used as the control for both larval and adult contact chemosensilla. 10 mM sinigrin and 10 mM nicotine were used as a positive control (induced the response of a bitter cell in the same sensilla) for larval and adult contact chemosensilla, respectively. All stimuli were tested randomly, and 12–20 replicates were set for each experiment.

## Effects of *Gr10* or *Gr6* knockout on behavioral responses of larvae to sugars and plant tissues

The feeding preference of WT, *Gr10*$^{-/-}$, and *Gr6*$^{-/-}$ larvae to 10 mM and 100 mM sucrose was investigated in two-choice tests as described earlier. 20 replicates were set for each experiment. We also performed the no-choice tests to measure the feeding amount of WT, *Gr10*$^{-/-}$, and *Gr6*$^{-/-}$ larvae in 1 hr

on leaves of cabbage (*Brassica oleraceaas*, Shenglv-7), corn kernels of maize (*Zea mays,* Jingke-968), seeds of pea (*Pisum sativum*, Changshouren), fruits of pepper (*Capsicum frutescens*, Changjian), and fruits of tomato (*Solanum lycopersicum*, Ailsa Craig-LA2838A). Fifth instar larvae on the first day after molting were starved for 2.5 hr before testing. Each plant tissue was weighed and put into a Petri dish (10 cm diameter ×2 cm depth; Corning, Wujiang, China), and then one larva was introduced. The larva was removed after 1 hr, and the rest of the plant tissue was weighed. Petri dishes with plant tissues but no caterpillar was used to measure water evaporation of the plant tissues. The feeding amount of larvae is calculated based on the weight changes of the plant tissue before and after being fed. Twenty replicates were set for each experiment.

## Effects of *Gr10* or *Gr6* knockout on behavioral responses of adults to sugars

The behavioral responses of the female adults of WT, $Gr10^{-/-}$, and $Gr6^{-/-}$ to sugars were determined by using the PER test as described earlier. sucrose, fucose, and fructose at the concentrations of 10 mM, 100 mM, and 1000 mM were tested randomly to antennae and tarsi. $H_2O$ was used as the control. All the experiments were repeated three times, and 20 individual moths were used for each set of tests.

## Statistical analysis

Statistical analyses were performed by SPSS 20 (IBM, Chicago, IL, USA) and GraphPad Prism 8.2.1 (Dotmatics, San Diego, CA, USA). For the tip-recording data and the PER data, the comparison between test compounds and the control was performed using two-tailed independent-samples *t* test. For the two-choice feeding test, paired-samples *t* test was used. The data of dose-response curves in tip-recording, PER experiments and two-electrode voltage-clamp recording, and gene relative expression levels and feeding amount in the no-choice test were analyzed by one-way ANOVA with Tukey's HSD tests. The comparison among mutation strains and WT in electrophysiological and PER experiments were analyzed by two-way ANOVA with post hoc Tukey's multiple comparison. All analytical methods were tested at p<0.05 for significant differences.

## Acknowledgements

We thank our colleagues Zi-Lin Li, Qing Zhang, Yan Chen, Jun Yang, Xin-Lin Gong, Yan-Yan Jia for assistances in PER experiments, mutant line screening and scanning electron microscopy. This work was supported by National Key R&D Program of China (Grant No. 2022YFD1400800), the National Natural Science Foundation of China (Grant No. 32130090), and the Beijing Natural Science Foundation (Grant No. 6224062).

## Additional information

### Funding

| Funder | Grant reference number | Author |
| --- | --- | --- |
| National Key R&D Program of China | 2022YFD1400800 | Chen-Zhu Wang |
| National Natural Science Foundation of China | 32130090 | Chen-Zhu Wang |
| Beijing Natural Science Foundation | 6224062 | Ke Yang |

The funders had no role in study design, data collection and interpretation, or the decision to submit the work for publication.

### Author contributions

Shuai-Shuai Zhang, Conceptualization, Data curation, Software, Formal analysis, Validation, Investigation, Visualization, Methodology, Writing – original draft, Writing – review and editing; Pei-Chao Wang, Resources, Data curation, Software, Investigation, Methodology; Chao Ning, Guo-Cheng Li,

Data curation, Software, Formal analysis, Methodology; Ke Yang, Funding acquisition, Investigation; Lin-Lin Cao, Investigation; Ling-Qiao Huang, Resources, Investigation, Methodology, Project administration; Chen-Zhu Wang, Conceptualization, Resources, Data curation, Software, Formal analysis, Supervision, Funding acquisition, Validation, Investigation, Visualization, Methodology, Writing – original draft, Project administration, Writing – review and editing

Author ORCIDs
Shuai-Shuai Zhang (iD) https://orcid.org/0000-0003-1163-9240
Ke Yang (iD) http://orcid.org/0000-0002-4138-3373
Chen-Zhu Wang (iD) http://orcid.org/0000-0003-0418-8621

Ethics
All experiments were approved by the Animal Care and Use Committee of the Institute of Zoology, Chinese Academy of Sciences and followed the Guide for the Care and Use of Laboratory Animals (IOZ17090-A).

Reviewer #1 (Public review): https://doi.org/10.7554/eLife.91711.3.sa1
Reviewer #2 (Public review): https://doi.org/10.7554/eLife.91711.3.sa2
Author response https://doi.org/10.7554/eLife.91711.3.sa3

## Additional files

### Supplementary files

- Supplementary file 1. GenBank accession numbers for sugar gustatory receptors used in this study.
- Supplementary file 2. The primer sequences used in PCR for full-length cloning of GRs. F: forward strand; R: reverse strand.
- Supplementary file 3. The primer sequences used in qRT-PCR for GR gene expression quantification. F: forward strand; R: reverse strand.
- Supplementary file 4. The primer sequences used in cDNA synthesis for *Xenopus oocytes* expression system. The italic sequences are protective bases, underline sequences are restriction enzymes, bold sequences are Kozak sequences. F: forward strand; R: reverse strand.
- Supplementary file 5. The primer sequences used in the experiments of Gr10 and Gr6 mutants establishment. F: forward strand; R: reverse strand.
- Supplementary file 6. Summary of the CRISPR/Cas9 directed mutation rates from G0 to G2 in the establishment of Gr10 and Gr6 mutants.
- MDAR checklist

### Data availability

Sequencing data have been deposited in GenBank under accession codes OP251144, OP251145, OP251146,OP251147,OP251148, OP251149, OP251150, and OP251151. All data generated or analysed during this study are included in the manuscript and supporting files; source data files have been provided for Figures 1–8.

The following datasets were generated:

| Author(s) | Year | Dataset title | Dataset URL | Database and Identifier |
|---|---|---|---|---|
| Zhang SS, Wang PC, Ning C, Yang K, Li GC, Cao LL, Huang LQ, Wang CZ | 2024 | The sucrose taste receptor of caterpillar is different from that of moth | https://www.ncbi.nlm.nih.gov/nuccore/OP251144 | NCBI GenBank, OP251144 |
| Zhang SS, Wang PC, Ning C, Yang K, Li GC, Cao LL, Huang LQ, Wang CZ | 2024 | The sucrose taste receptor of caterpillar is different from that of moth | https://www.ncbi.nlm.nih.gov/nuccore/OP251145 | NCBI GenBank, OP251145 |

*Continued on next page*

*Continued*

| Author(s) | Year | Dataset title | Dataset URL | Database and Identifier |
|---|---|---|---|---|
| Zhang SS, Wang PC, Ning C, Yang K, Li GC, Cao LL, Huang LQ, Wang CZ | 2024 | The sucrose taste receptor of caterpillar is different from that of moth | https://www.ncbi.nlm.nih.gov/nuccore/OP251146 | NCBI GenBank, OP251146 |
| Zhang SS, Wang PC, Ning C, Yang K, Li GC, Cao LL, Huang LQ, Wang CZ | 2024 | The sucrose taste receptor of caterpillar is different from that of moth | https://www.ncbi.nlm.nih.gov/nuccore/OP251147 | NCBI GenBank, OP251147 |
| Zhang SS, Wang PC, Ning C, Yang K, Li GC, Cao LL, Huang LQ, Wang CZ | 2024 | The sucrose taste receptor of caterpillar is different from that of moth | https://www.ncbi.nlm.nih.gov/nuccore/OP251148 | NCBI GenBank, OP251148 |
| Zhang SS, Wang PC, Ning C, Yang K, Li GC, Cao LL, Huang LQ, Wang CZ | 2024 | The sucrose taste receptor of caterpillar is different from that of moth | https://www.ncbi.nlm.nih.gov/nuccore/OP251149 | NCBI GenBank, OP251149 |
| Zhang SS, Wang PC, Ning C, Yang K, Li GC, Cao LL, Huang LQ, Wang CZ | 2024 | The sucrose taste receptor of caterpillar is different from that of moth | https://www.ncbi.nlm.nih.gov/nuccore/OP251150 | NCBI GenBank, OP251150 |
| Zhang SS, Wang PC, Ning C, Yang K, Li GC, Cao LL, Huang LQ, Wang CZ | 2024 | The sucrose taste receptor of caterpillar is different from that of moth | https://www.ncbi.nlm.nih.gov/nuccore/OP251151 | NCBI GenBank, OP251151 |

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

# Appendix 1

### Appendix 1—key resources table

| Reagent type (species) or resource | Designation | Source or reference | Identifiers | Additional information |
|---|---|---|---|---|
| Gene (Helicoverpa armigera) | Gr4 | GenBank | OP251144 | More details about this gene see *Supplementary file 1*. |
| Gene (H. armigera) | Gr5 | GenBank | OP251145 | More details about this gene see *Supplementary file 1* |
| Gene (H. armigera) | Gr6 | GenBank | OP251146 | More details about this gene see *Supplementary file 1* |
| Gene (H. armigera) | Gr7 | GenBank | OP251147 | More details about this gene see *Supplementary file 1* |
| Gene (H. armigera) | Gr8 | GenBank | OP251148 | More details about this gene see *Supplementary file 1* |
| Gene (H. armigera) | Gr9 | GenBank | XM_049843199 | More details about this gene see *Supplementary file 1* |
| Gene (H. armigera) | Gr10 | GenBank | OP251149 | More details about this gene see *Supplementary file 1* |
| Gene (H. armigera) | Gr11 | GenBank | OP251150 | More details about this gene see *Supplementary file 1* |
| Gene (H. armigera) | Gr12 | GenBank | OP251151 | More details about this gene see *Supplementary file 1* |
| Commercial assay or kit | RNeasy Plus Universal Mini Kit | Qiagen | Cat# 73404 | |
| Commercial assay or kit | Q5 High-Fidelity DNA Polymerase | NEB | Cat# M0491 | |
| Commercial assay or kit | TransStart FastPfu DNA Polymerase | TransGen Biotech | Cat# AP221-01 | |
| Commercial assay or kit | M-MLV reverse transcriptase | Promega | Cat# M1701 | |
| Commercial assay or kit | SYBR Premix Ex Taq | Takara | Cat# RR820 | |
| Commercial assay or kit | mMESSAGE mMACHINE SP6 | Ambion | Cat# AM1340 | |
| Commercial assay or kit | GeneArt gRNA Clean Up Kit | Invitrogen | Cat#A29377 | |
| Commercial assay or kit | GeneArt gRNA Prep Kit | Invitrogen | Cat#A29377 | |
| Commercial assay or kit | TrueCut Cas9 protein 2 | Invitrogen | Cat#A36498 | |
| Commercial assay or kit | Animal Tissue PCR Kit | TransGen Biotech | Cat#AD201-01 | |
| Chemical compound, drug | L - (+) - Arabinose | Sigma-Aldrich | CAS: 5328-37-0 | |
| Chemical compound, drug | D - (-) - Fructose | Sigma-Aldrich | CAS: 57-48-7 | |
| Chemical compound, drug | L - (-) - Fucose | Sigma-Aldrich | CAS: 2438-80-4 | |
| Chemical compound, drug | D - (+) - Galactose | Sigma-Aldrich | CAS: 59-23-4 | |
| Chemical compound, drug | D - (+) - Glucose | Sigma-Aldrich | CAS: 50-99-7 | |
| Chemical compound, drug | D - (+) - Mannose | Sigma-Aldrich | CAS: 3458-28-4 | |
| Chemical compound, drug | D - (+) - Xylose | Sigma-Aldrich | CAS: 58-86-6 | |
| Chemical compound, drug | D - Lactose monohydrate | Sigma-Aldrich | CAS: 64044-51-1 | |
| Chemical compound, drug | D - (+)-Maltose monohydrate | Sigma-Aldrich | CAS: 6363-53-7 | |
| Chemical compound, drug | Sucrose | Sigma-Aldrich | CAS: 57-50-1 | |
| Chemical compound, drug | D - (+) - Trehalose dihydrate | Sigma-Aldrich | CAS: 6138-23-4 | |

*Appendix 1 Continued on next page*

*Appendix 1 Continued*

| Reagent type (species) or resource | Designation | Source or reference | Identifiers | Additional information |
| --- | --- | --- | --- | --- |
| Chemical compound, drug | Sodium chloride | Sigma-Aldrich | CAS: 7647-14-5 | |
| Chemical compound, drug | Potassium chloride | Sigma-Aldrich | CAS: 7447-40-7 | |
| Chemical compound, drug | Magnesium chloride hexahydrate | Sigma-Aldrich | CAS: 7791-18-6 | |
| Chemical compound, drug | HEPES | Sigma-Aldrich | CAS: 7365-45-9 | |
| Software, algorithm | SAPID Tools software version 3.5 | *Smith et al., 1990*; | | |
| Software, algorithm | Autospike 3.7 | Syntech | | |
| Software, algorithm | MAFFT version 7.455 | *Rozewicki et al., 2019* | | |
| Software, algorithm | trimAl version 1.4 | *Capella-Gutiérrez et al., 2009*; | | |
| Software, algorithm | IQ-tree version 6.8 | *Nguyen et al., 2015* | http://iqtree.org/ | |
| Software, algorithm | pCLAMP software version 10.4.2.0 | Axon Instruments Inc | RRID:SCR_011323 | |
| Software, algorithm | SnapGene software version 4.3.8 | Insightful Science | https://www.snapgene.com/ | |
| Software, algorithm | SeqMan software version 7.1 | DNASTAR | https://www.dnastar.com/ | |
| Software, algorithm | SPSS 20 | IBM | https://www.ibm.com | |
| Software, algorithm | GraphPad Prism 8.2.1 | Dotmatics | https://www.graphpad.com/ | |
| Other | Primers for full-length cloning of GRs | This paper | | see *Supplementary file 2* |
| Other | Primers for qRT-PCR | This paper | | see *Supplementary file 3* |
| Other | Primers for Xenopus oocytes expression system | This paper | | see *Supplementary file 4* |
| Other | Primers for experiments of mutant strains establishment | This paper | | see *Supplementary file 5* |

