## [Editor Report · eLife assessment]

This **important** study identifies the gustatory receptors for sugar sensing in the larval and adult forms of the cotton bollworm, which is responsible for the destruction of many food crops world-wide. The authors find that the larval and adult forms utilise different receptors to sense sugars. The data are **convincing** and will be of interest neuroscientists working in sensory coding of sugars and to the pest management field.

---

## [Referee Report · Reviewer #1 (Public review)]

Summary:

The process of taste perception is significantly more intricate and complex in Lepidopteran insects. This investigation provides valuable insights into the role of Gustatory receptors and their dynamics in the sensation of sucrose, which serves as a crucial feeding cue for insects. The article highlights the differential sensitivity of Grs to sucrose and their involvement in feeding and insect behavior.

Strengths:

To support the notion of the differential specificity of Gr to sucrose, this study employed electrophysiology, ectopic expression of Grs in *Xenopus*, genome editing, and behavioral studies on insects. This investigation offers a fundamental understanding of the gustation process in lepidopteran insects and its regulation of feeding and other gustation-related physiological responses. This study holds significant importance in advancing our comprehension of lepidopteran insect biology, gustation, and feeding behavior.

Weaknesses:

While this manuscript demonstrates technical proficiency, there exists an opportunity for additional refinement to optimize comprehensibility for the intended audience. Several crucial sugars have been overlooked in the context of electrophysiology studies and should be incorporated. Furthermore, it is imperative to consider the potential off-target effects of Gr knock-out on other Gr expressions. This investigation focuses exclusively on Gr6 and Gr10, while neglecting a comprehensive narrative regarding other Grs involved in sucrose sensation.

---

## [Referee Report · Reviewer #2 (Public review)]

Summary:

To identify sugar receptors and assess the capacity of these genes the authors first set out to identify behavioral responses in larva and adult as well as physiological response. They used phylogenetics and gene expression (RNAseq) to identify candidates for sugar reception. Using first an in vitro oocyte system they assess the responses to distinct sugars. A subsequent genetic analysis shows that the Gr10 and Gr6 genes provide stage specific functions in sugar perception.

Strengths:

A clear strength of the manuscript is the breadth of techniques employed allowing a comprehensive study in a non-canonical model species.

Weaknesses:

There are no major weaknesses in the study for the current state of knowledge in this species. Since it is much basic work to establish a broader knowledge, context with other modalities remain unknown. It might have been possible to probe certain context known from the fruit fly, which would have strengthened the manuscript.

---

## [Author Response]

The following is the authors’ response to the original reviews.

We have revised the manuscript mainly in the following aspects: (1) the data of electrophysiological and behavioral responses of larvae and adults to trehalose have been added, and the related figures and texts have been modified accordingly; (2) the photos of taste organs of larvae and adults indicating the position of recorded sensilla have been added; (3) the potential off-target effects of GR knock-out on other GR expressions has been carefully explained and revised in the relevant text; (4) the abstract has been revised to present the findings more technically in a limited number of words; (5) some details of experiments in Materials and Methods and some new literatures have been added; (6) a new figure (Figure 8) summarizing the main findings of the study has been added.

In the following, we respond to the reviewers’ comments and suggestions one by one. We hope that our answers will satisfy you and the three reviewers. We are also very happy to get further valuable advices from you.

**Public Reviews:**

**Reviewer #1 (Public Review):**
Summary:The process of taste perception is significantly more intricate and complex in Lepidopteran insects. This investigation provides valuable insights into the role of Gustatory receptors and their dynamics in the sensation of sucrose, which serves as a crucial feeding cue for insects. The article highlights the differential sensitivity of Grs to sucrose and their involvement in feeding and insect behavior.Strengths:To support the notion of the differential specificity of Gr to sucrose, this study employed electrophysiology, ectopic expression of Grs in *Xenopus*, genome editing, and behavioral studies on insects. This investigation offers a fundamental understanding of the gustation process in lepidopteran insects and its regulation of feeding and other gustation-related physiological responses. This study holds significant importance in advancing our comprehension of lepidopteran insect biology, gustation, and feeding behavior.

Thank you for your recognition of our research.

Weaknesses:While this manuscript demonstrates technical proficiency, there exists an opportunity for additional refinement to optimize comprehensibility for the intended audience. Several crucial sugars have been overlooked in the context of electrophysiology studies and should be incorporated. Furthermore, it is imperative to consider the potential off-target effects of Gr knock-out on other Gr expressions. This investigation focuses exclusively on Gr6 and Gr10, while neglecting a comprehensive narrative regarding other Grs involved in sucrose sensation.

We accept the reviewer's suggestion. Because trehalose is a main sugar in insect blood, and it is converted by insects after feeding on plant sugars, we have added the new data on electrophysiological and behavioral responses of larvae and adults of Helicoverpa armigera to trehalose (see Figure 1-2, Figure 1-figure supplement 1, Figure 2-figure supplement 1). Now, the total eight sugars include 2 pentoses (arabinose and xylose), 4 hexoses (fructose, fucose, galactose and glucose), and 2 disaccharides (sucrose and trehalose), which were chosen because they are mainly present in host-plants of H. armigera and/or representative in the structure and source of sugars.

We fully agree to the reviewer’s opinion and have already taken the potential off-target effects of CRISPR/Cas9 knockout of Gr on other GR expressions into consideration. To predict the potential off-target sites of sgRNA of Gr6 and Gr10 establishing homozygous mutants using CRISPR/Cas9 technology, we first use online software CasOFFinder (http://www.rgenome.net/cas-offinder/) to blast the genome of the wild type cotton bollworm and set the mismatch number less than or equal to 3. We found that Gr10 sgRNA had no potential potential off-target site, and the sgRNA of Gr6 had only one potential off-target site. Therefore, we designed primers according to the sequence of potential off-target sites of Gr6 sgRNA, and conducted PCR using genomic DNA of homozygous mutant as a template， performed Sanger sequencing on the PCR products obtained, and found that the potential off-target sites of Gr6 sgRNA were no different from those of the wild type. Particularly, concerning the sgRNA of Gr6 and Gr10 may produce off-target effects on other sugar receptor genes of H. armigera, we conducted the same off-target site analysis with the designed sgRNA on each of the other eight sugar receptor genes, and found that there were no off-target sites on these receptor genes (see Line254-256).

**Reviewer #2 (Public Review):**
Summary:To identify sugar receptors and assess the capacity of these genes the authors first set out to identify behavioral responses in larvae and adults as well as physiological response. They used phylogenetics and gene expression (RNAseq) to identify candidates for sugar reception. Using first an in vitro oocyte system they assess the responses to distinct sugars. A subsequent genetic analysis shows that the Gr10 and Gr6 genes provide stage specific functions in sugar perception.Strengths:A clear strength of the manuscript is the breadth of techniques employed allowing a comprehensive study in a non-canonical model species.

Thank you for your recognition of our research.

Weaknesses:There are no major weaknesses in the study for the current state of knowledge in this species. Since it is much basic work to establish a broader knowledge, context with other modalities remains unknown. It might have been possible to probe certain contexts known from the fruit fly, which would have strengthened the manuscript.

Thank you so much for your suggestion. According to this suggestion, we further added some sentences probing sugar sensing and behaviors of fruit fly larvae in the Introduction and discussion sections (Line 68-71 in Introduction section, Line 395-399 in Discussion section).

**Reviewer #3 (Public Review):**
In this study, the authors combine electrophysiology, behavioural analyses, and genetic editing techniques on the cotton bollworm to identify the molecular basis of sugar sensing in this species.The larval and adult forms of this species feed on different plant parts. Larvae primarily consume leaves, which have relatively lower sugar concentrations, while adults feed on nectar, rich in sugar. Through a series of experiments-spanning electrophysiological recordings from both larval and adult sensillae, qPCR expression analysis of identified GRs from these sensillae, response profiles of these GRs to various sugars via heterologous expression in *Xenopus oocytes*, and evaluations of CRISPR mutants based on these parameters-the authors discovered that larvae and adults employ distinct GRs for sugar sensing. While the larva uses the highly sensitive GR10, the adult uses the less sensitive and broadly tuned GR6. This differential use of GRs are in keeping with their behavioral ecology.The data are cohesive and consistently align across the methodologies employed. They are also well presented and the manuscript is clearly written.
**Recommendations for the authors:**
While appreciating the quality of the work and its presentation, we have a few comments for the authors, should they wish to consider them, that would significantly improve the presentation of the work.Title: Could the authors please revisit their title to better reflect the main finding of their work?

The title has been changed into “The larva and adult of Helicoverpa armigera use differential gustatory receptors to sense sugars”.

Text: There are a few comments related to the text, and these are listed below:(1) Could the authors place their work in the context of what's known about sugar sensing in *Drosophila* larva and adult?

In the Introduction section, we added the status of research on sugar perception in *Drosophila* larvae, pointing out "No external sugar-sensing mechanism in *Drosophila* larvae has yet been characterized." (Line 70-71); in the Discussion section, the research progress of sugar sensing in *Drosophila* adults and larvae was also summarized (Line 397-399).

(2) For each results section, could the authors please include a sentence or two that interprets the data in the context of previously presented data?

We accept the reviewer's suggestion. In order to make it easy for readers to follow up, we included a sentence interprets the above data at the beginning of each part of the Results on the premise of avoiding duplication.

(3) Could the authors please provide details of the generation and screening of the CRISPR mutants?

We have added more details on mutant establishment and screening in the Materials and Methods section (Line 722-726, 729-732).

Figures: Could the authors please include images and schematics wherever possible? For example, a schematic depicting the position of the sense organs and one summarising the main findings of the studies.

In Figure 1 we added the photo of each taste organ, on which the recorded sensilla were indicated. We also added a new figure, Figure 8, summarizing the main findings of the study.

Choice of Sugars: Could the authors please justify their choice of sugars they have used in the analyses?

In the first paragraph of the Results section of the article, we further explain the reasons for using the sugars in the study. “We first investigated the electrophysiological responses of the lateral and medial sensilla styloconica in the larval maxillary galea to eight sugars. These sugars were chosen because they are mostly found in host-plants of H. armigera or are representative in the structure and source of sugars.”

In addition to this, there are several specific comments in the detailed reviewers comments below, which the authors could consider responding to.
**Reviewer #1 (Recommendations For The Authors):**
The article titled "Sucrose taste receptors exhibit dissimilarities between larval and adult stages of a moth" by Shuai-Shuai Zhang and colleagues provides an intriguing analysis. The authors have conducted a meticulously planned and executed study. However, I do have some inquiries.(1) What precisely does the term "differ" signify in the title? It can be expounded upon in terms of differing in expression or sensitivity. The title could benefit from being more informative. The authors should appropriately specify the insect species in the title of the paper. This would make it more comprehensible to readers. Merely mentioning the term "moth" does not provide any information about the model organism. Hence, it would be preferable to mention Helicoverpa armigera instead of using the generic term "moth" in the title.

Thank you for your suggestions. We considered it better to emphasize that the receptors for sucrose are different, and we have accepted the suggestion of adding the name of the animal. The title has been changed into “The larva and adult of Helicoverpa armigera use differential gustatory receptors to sense sucrose”.

(2) The abstract is written in a simple and easily understandable manner, but it overlooks important findings from a technical standpoint.

We add some key experimental techniques to illustrate some important findings in the Abstract.

(3). Almost all herbivorous insects are said to consume plants and utilize sucrose as a stimulus for feeding, as stated by the authors. Sucrose, glucose, and fructose sugar are among the commonly observed stimulants for feeding in numerous insects. It would be appropriate to incorporate not only sucrose but also glucose and fructose as feeding stimulants for almost all herbivorous insects.

Thank you for your suggestion. Sucrose is the major sugar in plants, and its concentration varies greatly from tissue to tissue, while the concentration of the hexose sugars is much lower and the concentration does not change much. In Line 48, we state that sucrose, glucose, and fructose are feeding stimuli for herbivorous insects. From the previous studies, it seems that sucrose is the strongest, followed by fructose, and finally glucose. The cotton bollworm larvae showed no electrophysiological and behavioral response to glucose.

(4) The reason why trehalose is not considered in the electrophysiology analysis is unclear. Given that trehalose is a major sugar in insects and plants, it would be intriguing to include it in the analysis.

We have accepted the reviewer's suggestion, and supplemented the electrophysiological responses of taste organs in larvae and adults of Helicoverpa armigera to trehalose (Figure 1, Figure 1-Figure Supplement 1), and also tested the behavioral responses of the larvae and adults to trehalose (Figure 2, Figure 2-Figure Supplement 1). Therefore, all the related figures have been changed.

(5) The author's intention regarding the co-receptor relationship between Gr5 and Gr6 (line 211) is unclear. If this is indeed the case, then the reason for considering Gr5 in further studies remains uncertain.

We have changed the sentence as follows: “Since Gr5 was highly expressed with Gr6 in the proboscis and tarsi (Figure 3D-3E, Figure 3—figure supplement 1), we suspected that Gr5 and Gr6 might be expressed in the same cells, and then tested the response profile of their co-expression in oocytes.”

(6) The homologous nature of Grs is emphasized by the authors. It is not specified how the author ensured that the guide RNA targeting Gr6 or Gr10 did not result in off-target effects on other Grs.

Thank you so much for your suggestion. We have rewritten the relevant paragraph (Line 238-251), detailing our tests and the results on the potential off-target effects of knocking out GRs by CRISPR/Cas9: “In order to predict the potential off-target sites of sgRNA of Gr6 and Gr10, we used online software Cas-OFFinder (http://www.rgenome.net/cas-offinder/) to blast the genome of H. armigera, and the mismatch number was set to less than or equal to 3. According to the predicted results, the Gr10 sgRNA had no potential off-target region but Gr6 sgRNA had one. Therefore, we amplified and sequenced the potential off-target region of Gr6-/- and found there was no frameshift or premature stop codon in the region compared to WT (Figure 5—figure supplement 2). It is worth mentioning that there was no potential off-target region of Gr6 and Gr10 sgRNA in other sugar receptor genes of H. armigera, Gr4, Gr5, Gr7, Gr8, Gr9, Gr11 and Gr12. We further found there was no difference in the response to xylose of the medial sensilla styloconica among WT, Gr10-/- and Gr6-/- (Figure 5—figure supplement 2). Furthermore, WT, Gr10-/- and Gr6-/- did not show differences in the larval body weight, adult lifespan, and number of eggs laid per female (Figure 5—figure supplement 2). All these results suggest that no off-target effects occurred in the study.”

(7) Is it possible that knocking out Gr10 is not compensated for by the overexpression of Gr6 or other sucrose sensing Grs? Similarly, would the vice versa scenario hold true?

In the Discussion section, we have added some sentences to discuss this issue: “From our results, knocking out Gr10 or Gr6 is unlikely to be compensated by overexpression of other sugar GRs. One of our recent studies showed that Orco knockout had no significant effect on the expression of most OR, IR and GR genes in adult antennae of H. armigera, but some genes were up- or down-regulated (Fan et al., 2022).”

(8) What was the rationale for selecting nine candidate GR genes for expression analysis?

Based on the reviewer's suggestion, we expanded the relevant paragraphs to illustrate the rationale for selecting nine candidate GR genes for expression analysis: “To reveal the molecular basis of sugar reception in the taste sensilla of H. armigera, we first analyzed the putative sugar gustatory receptor genes based on the reported gene sequences of GRs in H. armigera and their phylogenetic relationship of *D. melanogaster* sugar gustatory receptors (Jiang et al., 2015; Pearce et al., 2017; Xu et al., 2017). Nine putative sugar GR genes, Gr4–12 were identified, and their full-length cDNA sequences were cloned (The GenBank accession number is provided in Appendix—Table S1).” (Line 155-161)

(9) What is the potential reason for the difference between the major larval sugar receptors of *Drosophila* and Lepidopterans?

The difference between the major larval sugar receptors of *Drosophila* and Lepidopterans is probably due to differences in the food their larvae feed on. Fruit fly larvae feed on rotten fruit, the main sugar of which is fructose. The larvae of Lepidoptera mainly feed on plants, and the main sugar is sucrose. In the Discussion section, we have added a sentence “This is most likely due to fruit fly larvae feeding on rotten fruits, which contain fructose as the main sugar.” (Line 399-401)

(10) There is a disparity in GRs, specifically GR5 and GR6, between the female antenna, proboscis, and tarsi. What could be the possible justification and significance of this?

Thank you so much for this question. We have added a sentence in the Discussion section, “In this study, the expression patterns of 9 sugar GRs in three taste organs of adult H. armigera show that there is a disparity in GRs, specifically GR5 and GR6, between the female antenna, tarsi and proboscis, which may be an evolutionary adaptation reflecting subtle differentiation in the function of these taste organs in adult foraging. Antennae and tarsi play a role in the exploration of potential sugar sources, while the proboscis plays a more precise role in the final decision to feed.” (Line 433-438)

(11) I suggest that a visual representation illustrating the positioning of GSNs, particularly the lateral and medial sensilla, in both larva and adult stages would enhance the correlation with the results.

In Figure 1 we added the photo of each taste organ and the position of the recorded sensilla, and also added a new figure, Figure 8 summarizing the main findings of the studies.

(12) Further experiments can be conducted to elucidate the precise molecular mechanisms, particularly the downstream effects of GRs, in order to establish the specificity of GRs more convincingly.

Thank you so much for your suggestion. We have discussed the further experiments in the Discussion section, “To elucidate the precise molecular mechanisms of sugar reception in H. armigera is necessary to compare a series of single, double and even multiple Gr knock-out lines and investigate the downstream effects of the GRs.” (Line 363-369)

(13) Figure 6 caption: In Figure 6 (D to I), the percentage of PER is depicted. There is redundancy in the Y-axis title (Percentage of PER) and the legend. This appears to be repetitive. I suggest that it would be better to include the Y-axis title only in Figure D or in Figures D and G.

We accept the suggestion. Figure 7 (not Figure 6) has been revised accordingly.

(14) In Figures 6A and 6C, there is inconsistency in the colors used for WT, Gr6, and Gr10. This could potentially confuse the reader. I recommend using the same colors in both figures instead of using a blue color. Please specify how the authors calculated the feeding area in Figure 6.

We accept the reviewer's suggestion and have changed the color of Figure 7A, B. We have also added the detail method for calculating feeding area (Line 541-545).

(15) In Two-choice tests, why did the authors use 0.01% Tween 80? Please provide comments on this.

Use of 0.01% Tween 80 is to reduce the surface tension and increase the malleability of the solution. We have given detailed explanation in the Method section and cite the reference. (Line538-540)

(16) It would be valuable if the authors could comment on the prospects of this study, considering that GRs play a vital role in controlling behavior and developmental pathways. What are the potential consequences of blocking or disrupting these receptors in terms of behavioral and developmental phenotypic deformities? Could this potentially lead to increased insect mortality?

Thank you so much for your suggestions. In the last paragraph of the Discussion section, we have added the following perspectives, “Knockout of Gr10 or Gr6 led to a significant decrease in sugar sensitivity and food preference of the larvae and adults of H. armigera, respectively, which is bound to bring adverse consequences to survival and reproduction of the insects. Therefore, studying the molecular mechanisms underlying sugar perception in phytophagous insects may provide new insights into the behavioral ecology of this important and highly diverse group of insects, and measures blocking or disrupting sugar receptors could also have applications to control agricultural pests and improve crop yields worldwide” (Line 449-456).

**Reviewer #2 (Recommendations for The Authors):**
There are a few comments, that I feel would be beneficial to be addressed.The authors used 7 different sugars for their experimental approach. While I agree that this is a sufficiently large collection for a study, I was wondering why they specifically chose these sugars; an explanatory section might be helpful for a reader to follow the reasoning.

According to reviewer 1's suggestion, we increased trehalose to 8 sugars in experiments. Trehalose is a main sugar in insect blood. It is converted by insects after feeding on plant sugars. The 8 sugars were chosen because they are present in host-plants of H. armigera or are representative in the structure and source of sugars. They contain 2 pentoses (arabinose and xylose), 4 hexoses (fructose, fucose, galactose and glucose), and 2 disaccharides (sucrose and trehalose).

It might be beneficial to provide some broader overview on the gustatory system in the cotton bollworm, particularly at the larval stage since this may not be common knowledge. Along these lines eg. the complexity of sensilla types, organs and overall number (or estimation) of neurons might be good to know, a graphical representation of the sense organs might be informative.

In the Introduction section, we give a more specific description on sugar sensitive GSNs in the taste system of the larva and adult of H. armigera, and cite the corresponding references.

Concerning phylogeny of GRs, it might be relevant to know how complete the genome information is and some more general background on GR diversity in the cotton bollworm.

We agree to your opinion. According to this idea, we got the putative sugar GRs from the previously published genome (Pearce et al. 2017) and the related annotation of GRs (Jiang et al. 2015, Xu et al. 2012). We have made a more detailed explanation about this in the new version of the manuscript, “We first analyzed the putative sugar gustatory receptor genes based on the genome data of H. armigera (Pearce et al. 2017), the reported gene sequences of sugar GRs in H. armigera and their phylogenetic relationship of *D. melanogaster* sugar gustatory receptors (Jiang et al. 2015, Xu et al. 2012). All nine putative sugar GR genes in H. armigera, Gr4–12 were validated, and their full-length cDNA sequences were cloned (The GenBank accession number is provided in Appendix—Table S1).” (Line 155-161).

Generation of mutants based on CRISPR is intriguing and a powerful step. While the techniques are well described in the method section, there is no information concerning efficiency or broader feasibility of the approach. I feel it would be quite interesting to learn about how feasible or laborious the approach is to generate mutants (e.g. number of initial injected eggs, the resulting F0 offspring, number of back-crosses, number of screened F1s ....).

In the Materials and Methods section, we have added specific success rates for each step in the process of building the two mutants (Line 722-726, 729-732).

**Reviewer #3 (Recommendations For The Authors):**
I want to congratulate the authors on this very nice study and have only minor comments for them.(1) It would be very nice to include pictures of the larva and adult of H. armigera. It would also help to have schematics of where the sensilla they are recording from are.

We have added photos of four taste organs on which the recoded sensilla were indicated (Figure 1), and picture of the larva and adult on which the stimulating site was indicated (Figure 2).

(2) A schematic summarising their findings, including the relevance to the animal's behavioural ecology, will greatly improve interpretations for the broader audience.

A schematic summarizing the findings has been added.

(3) The manner in which PIs are represented in figure 2A, B (among others) is confusing. Can the authors please plot the PI and not the feeding area? From the PI values listed beside the plot, it actually suggests that the larvae don't really show a preference. Could the authors please comment on this?

Yes, sucrose has a significant stimulating effect on larva feeding, but the effect is not as large as the predicted based on the sensitivity of the sensillum, the main reasons are as follows: (1) there are many factors affecting larva feeding, sucrose is only one of them; (2) due to the substrate leaf discs also contain sugar, the effect of newly added sucrose may be reduced. After careful consideration, we think it is better to display the feeding area and PI together so that readers have a complete understanding of the data.

(4) The heterologous expression experiments suggest that co-expression of GR6 with either GR10 or GR5 somehow suppress the response of the GR6 alone to fucose. Am I reading the data correctly? Why would this be? Perhaps the authors could discuss this. In this context, it would help to reproduce all the GR6 data together.

Your interpretation is reasonable to a certain extent. The result of co-injection might be that Gr10 or Gr5 inhibited the response of Gr6. However, there is another possibility that the amount of Gr6 sRNA was diluted by co-injection of two GRs, resulting in a reduced response of Gr6 to fucose.

(5) In general, for each results section, it would help to have a sentence or two that interprets the data in the context of previously presented data. This would help the reader digest the data and interpret it as they read along. Currently, the authors summarise the observations and leave all the interpretation to the discussion section.

We accept the suggestion. In each part of the results, we have added a sentence to explain the above data, which will help readers to clarify the context of the research more easily.

(6) Is the GR6 data in 4C not lined up correctly?

Yes, it is right.

(7) Line 228 suggests that the mutants were validating with qPCRs - I don't see that data.

The mutants were not validating with qPCR. We used the ordinary PCR technology at the mRNA level to verify whether the related sequences were really deleted in the mutants.